# Little skate genome provides insights into genetic programs essential for limb-based locomotion

DongAhn Yoo[1†], Junhee Park[2†], Chul Lee[1], Injun Song[2], Young Ho Lee[1], Tery Yun[2], Hyemin Lee[3], Adriana Heguy[4], Jae Yong Han[5], Jeremy S Dasen[6*], Heebal Kim[1,7,8*], Myungin Baek[2*]

[1]Interdisciplinary Program in Bioinformatics, Seoul National University, Seoul, Republic of Korea; [2]Department of Brain Sciences, DGIST, Daegu, Republic of Korea; [3]Department of Biology, Graduate School of Arts and Science, NYU, New York, United States; [4]Genome Technology Center, Division for Advanced Research Technologies, and Department of Pathology, NYU School of Medicine, New York, United States; [5]Department of Agricultural Biotechnology, Seoul National University, Seoul, Republic of Korea; [6]Neuroscience Institute, Department of Neuroscience and Physiology, New York University School of Medicine, New York, United States; [7]Department of Agricultural Biotechnology and Research Institute of Agriculture and Life Sciences, Seoul National University, Seoul, Republic of Korea; [8]eGnome, Inc, Seoul, Republic of Korea

*For correspondence:
Jeremy.Dasen@nyumc.org (JSD);
heebal@snu.ac.kr (HK);
bmi008@dgist.ac.kr (MB)

[†]These authors contributed equally to this work

[‡]Is an employee of eGnome, Inc

**Abstract** The little skate *Leucoraja erinacea*, a cartilaginous fish, displays pelvic fin driven walking-like behavior using genetic programs and neuronal subtypes similar to those of land vertebrates. However, mechanistic studies on little skate motor circuit development have been limited, due to a lack of high-quality reference genome. Here, we generated an assembly of the little skate genome, with precise gene annotation and structures, which allowed post-genome analysis of spinal motor neurons (MNs) essential for locomotion. Through interspecies comparison of mouse, skate and chicken MN transcriptomes, shared and divergent gene expression profiles were identified. Comparison of accessible chromatin regions between mouse and skate MNs predicted shared transcription factor (TF) motifs with divergent ones, which could be used for achieving differential regulation of MN-expressed genes. A greater number of TF motif predictions were observed in MN-expressed genes in mouse than in little skate. These findings suggest conserved and divergent molecular mechanisms controlling MN development of vertebrates during evolution, which might contribute to intricate gene regulatory networks in the emergence of a more sophisticated motor system in tetrapods.

## Editor's evaluation

This study provides the genome of the little skate *Leucoraja erinacea*, a cartilaginous fish that displays pelvic fin-driven walking-like behavior. Leveraging this genomic resource, the authors compare gene expression and chromatin accessibility profiles in motor neurons of the little skate and other species (e.g., mouse, chicken), aiming to predict conserved and divergent gene regulatory mechanisms underlying motor neuron development. The work represents an important contribution to the field of comparative genomics and evolutionary biology.

## Introduction

The little skate *Leucoraja erinacea* is a marine species of jawed vertebrate which shares a common ancestor with tetrapods 473 MYA (*Kumar et al., 2017*). Yet, it displays a walking-like behavior which resembles limb-based locomotion (*Holst and Bone, 1993*; *Koester et al., 2003*; *Jung et al., 2018*), suggesting that the genetic programs and neural subtypes essential for walking existed in a common ancestor of cartilaginous fish and tetrapods. Hence, *Leucoraja erinacea* is a useful model organism for the study of locomotor circuits, as it may provide insights into the genetic programs underlying limb-driven locomotion that may have originated in the common ancestor of vertebrates with paired appendages (*Gillis and Shubin, 2009*). Previous developmental studies on the little skate demonstrated a conserved Hox TF-dependent regulatory network that specifies MNs innervating fin and limb muscle (*Jung et al., 2018*). Little skate pectoral and pelvic fin MNs express forelimb MN Hox genes, *Hoxa6/7* and a hindlimb MN Hox gene, *Hoxa10*, respectively. In addition, the little skate genome lacks the *Hoxc* cluster (*King et al., 2011*) and its spinal cord is occupied by fin MNs without an inter-fin region, which is reminiscent of the *Hoxc* cluster mutant mouse (*Jung et al., 2014*). However, the mechanisms that regulate expression of genes controlling motor circuit development remain to be identified, and how different species generate MN subtypes to innervate ~10 pelvic fin muscles of little skate (*Macesic and Kajiura, 2010*) versus ~50 limb muscles of tetrapods (*Sweeney et al., 2018*; *Landmesser, 1978*; *Sullivan, 1962*) remains to be determined.

To investigate common and divergent regulatory gene networks of the little skate and tetrapods, integrated analysis of comparative genomics, transcriptomics and epigenomics analyses, including assay for transposase-accessible chromatin using sequencing (ATAC-seq) can be applied. However, in little skate, such genome-wide omics studies have been limited due to severe fragmentation of the reference genome (*Wyffels et al., 2014*). Limitations of short-read sequencing, such as the 500 bp read length in the case of skate genome, are unable to cover long repeats, leading to poor assembly of repeat-rich regions. Continuous long read (CLR) sequencing provides a solution to such assembly errors. With the assistance of available high-quality reference assemblies of phylogenetically close species, long read sequencing allows for reference-guided scaffolding and comparative annotations approaches (*Alonge et al., 2019*; *Fiddes et al., 2018*).

In response to the limited reference genome of little skate, this and a similar study by Marletaz et al. generated new genome assemblies (*Marletaz et al., 2022*). Marletaz et al. generated a new chromosome-scale genome via combination of Pacbio, Illumina and Hi-C sequencing. Here, we generated a new genome assembly of 2.13 Gb and gene annotation of little skate by applying a combination of PacBio long-read and Illumina short-read sequencing analyses, a state-of-art assembly and annotation pipeline (*Alonge et al., 2019*; *Fiddes et al., 2018*; *Kolmogorov et al., 2019*). Based on the new reference genome, transcriptome and ATAC-seq analyses were performed to identify gene expression patterns and chromatin accessibility of fin-innervating MNs. Through a comparative transcriptome analysis with two well-studied tetrapods (mouse and chick), we identified both conserved and divergent gene expression patterns among different species. Comparative chromatin accessibility analysis revealed more TF motif predictions in the MN-expressed genes in mouse than in little skate, which might support the emergence of a more intricately regulated motor system during evolution. The findings of this study provide deeper insights into the origin of genetic programs underlying limb-based locomotion.

## Results

### High-quality little skate genome assembly through a combination of long and short read sequencing

The genome of *Leucoraja erinacea* was assembled using a combination of PacBio long read and Illumina short read data (*Figure 1A and B*). Presented in *Figure 1A* are the top 49 scaffolds ranked by the length which are equivalent to the number of chromosomes in thorny skate genome (sAmbRad1) (*Rhie et al., 2021*). The 49 scaffolds constituted 97.8% of the little skate genome. Compared to the previous assembly, our assembled genome size increased by 36.5%–2.13 Gb (*Wyffels et al., 2014*) ~93% of the estimated genome size based on K-mer spectra of whole genome sequencing data (*Figure 1—figure supplement 1*). The contiguity of the genome was also improved by over 300-fold, in terms of contig N50 of 214 Kb. The completeness of the new genome assembly is highlighted in

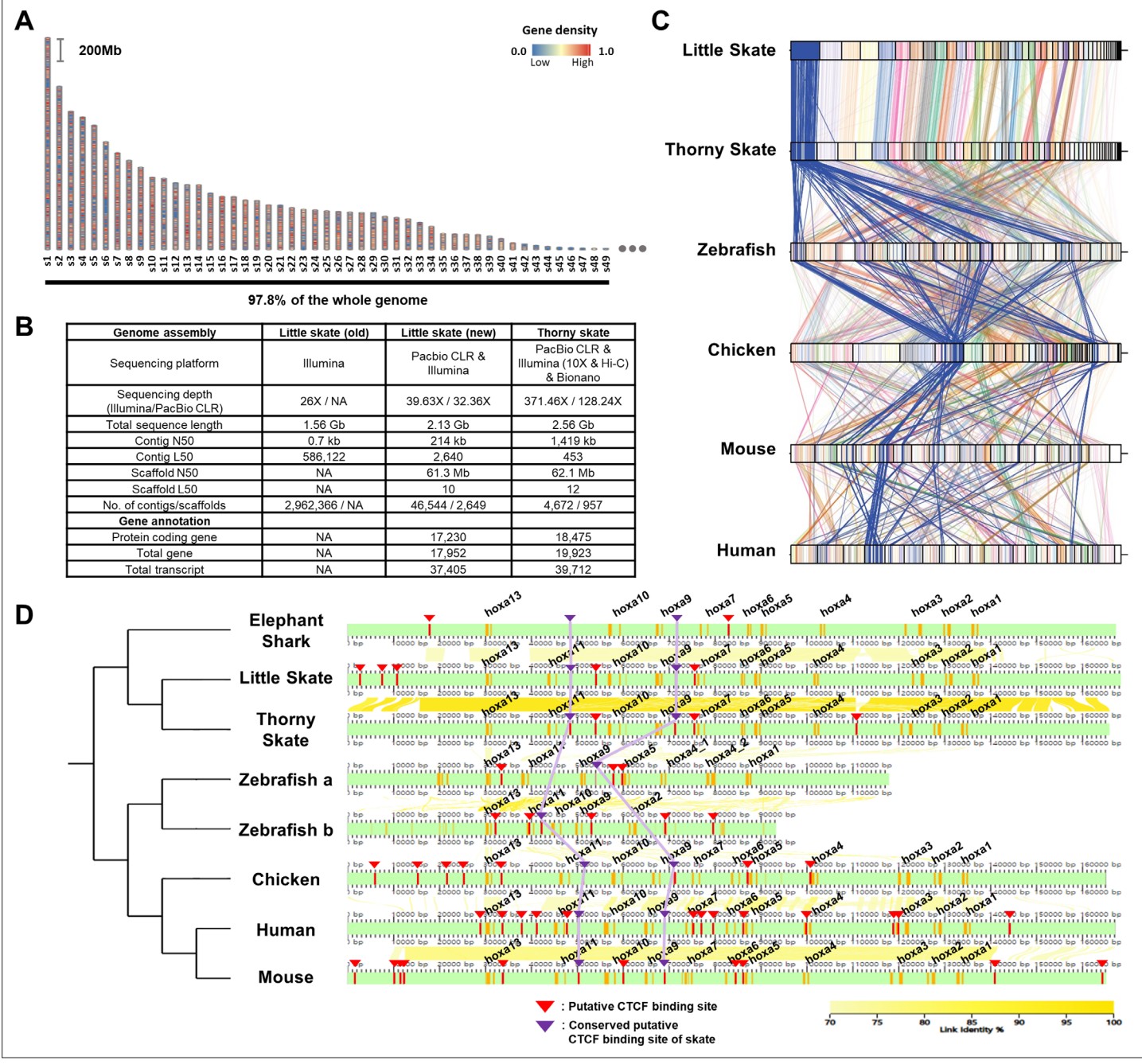

**Figure 1.** Assembly and comparison of little skate genomes across diverse phylogenetic lineages. (**A**) Top 49 scaffolds ranked by the length. Gene density in the scaffolds is color coded. (**B**) Assembly statistics of the skate genome. (**C**). Comparison of BUSCO gene synteny across diverse phylogenetic lineages. The BUSCO genes in each scaffold of the new little skate genome are color coded and the longest scaffold (**s1**) is highlighted in blue. (**D**) Putative motifs of CTCF in the *Hoxa* cluster. CTCF motifs are predicted (described in materials and methods) and indicated by red bars and arrowheads and the ones conserved by purple bars and arrowheads. The exons in the genome are denoted by orange bars. The sequence identities of alignment blocks are indicated by the intensity of yellow color.

The online version of this article includes the following figure supplement(s) for figure 1:

**Figure supplement 1.** Genome size estimation via Genome Scope 2.

**Figure supplement 2.** Genome assembly process.

**Figure supplement 3.** BUSCO genes and repetitive elements of little skate genome.

**Figure supplement 4.** Limitation of previous transcriptome data.

**Figure supplement 5.** Qualitative assessment of genome assembly of *Hoxa* cluster.

*Figure 1 continued on next page*

**Figure supplement 6.** Comparison of *Hoxc* clusters.

the benchmarking universal single-copy orthologs (BUSCO) gene contents (*Figure 1—figure supplements 2–3*). Missing and fragmented BUSCO which was 69.1% and 23.9% in the previous assembly has been reduced to 7.3% and 3.6% in our assembly, respectively. Also, the proportion of complete BUSCO genes (89.1%) as well as the long transcript lengths demonstrate improved completeness than that of the previous assembly (*Figure 1—figure supplements 3 and 4*). As repetitive elements are reported to have large variation among cartilaginous fish genomes (*Rhie et al., 2021*; *Hara et al., 2018*), the repeats were investigated using Repeatmasker (*Smit et al., 2013*) and were compared. Approximately, 63.6% of the little skate genome is occupied by repetitive elements (*Figure 1—figure supplement 3B*), which may account for its relatively large genome size compared to elephant shark and zebrafish genomes (C.milii-6.1.3 and GRCz11) (*Venkatesh et al., 2014*; *Howe et al., 2013*). Among the repeats, long interspersed nuclear elements (LINE) were the most abundant of all classes which is similar to thorny skate and three shark genome assemblies (*Hara et al., 2018*).

A scaffold N50 of 61.3 Mb of our little skate genome assembly was comparable to a recently published high-quality thorny skate genome assembly (*Rhie et al., 2021*; 62.1 Mb; *Figure 1B*). The number of protein-coding genes (17,230) and transcripts (17,952) predicted in the new genome assembly was similar to the number in the thorny skate genome (*Figure 1B*). As a qualitative assessment of our genome assembly, we compared an existing sequence of the *Hoxa* cluster constructed using a BAC clone (*Mulley et al., 2009*) (FJ944024), to our assembled genome. Coding sequences in our assembled *Hoxa* cluster aligned well with the fully sequenced BAC clone (*Figure 1—figure supplement 5*), highlighting the reliability of the new assembly. We observed complete loss of *Hoxc* cluster in the little skate genome which was also shown in the thorny skate genome (*King et al., 2011*; *Rhie et al., 2021*; *Criswell et al., 2021*) and relatively high repeat contents nearby genomic regions in the scaffold41 (s41), which contains the largest number of the *Hoxc* neighbor genes (*Figure 1—figure supplement 6*). Although a residual fragment of *Hoxc* cluster was reported in the shark genomes (*Hara et al., 2018*), cloudy cat shark, bamboo shark, and whale shark genomes are not visualized here due to an absence of gene annotation files.

Comparing the new little skate genome with the one of thorny skate, we found high level of conservation in terms of BUSCO gene contents (*Figure 1C*). Interestingly, we also observed that the BUSCO gene synteny of little skate showed more similar patterns with chicken and mammals than with zebrafish (*Figure 1C*). In addition to gene content, we also compared putative CTCF motifs, which are known to regulate expression of *Hoxa* genes during development (*Kim et al., 2011*; *Narendra et al., 2015*; *Figure 1D*). In tetrapods, *Hoxa* genes are expressed in spinal cord MNs and specify their subtype identities (*Jung et al., 2014*; *Dasen et al., 2005*; *Lacombe et al., 2013*), and similarly in little skate, *Hoxa* genes are expressed in spinal cord MN subtypes (*Jung et al., 2018*). Among previously reported CTCF sites in mouse (*Narendra et al., 2015*), one located between *Hoxa7* and *Hoxa9*, and between *Hoxa10* and *Hoxa11* were conserved across diverse species and were also observed in little skate *Hoxa* cluster despite the phylogenetic distance (*Figure 1D*). On the other hand, the CTCF motifs near *Hoxa13* upstream and between *Hoxa5* and *Hoxa6* which are conserved in other species were not found in putative CTCF sites of skate, elephant shark, and other sharks reported previously (*Hara et al., 2018*). The absence of the CTCF site between *Hoxa5* and *Hoxa6* may contribute to the relatively expanded domain of *Hoxa9* in fin innervating MNs (*Jung et al., 2018*; *Narendra et al., 2015*).

## DEG analysis with a new reference genome revealed comprehensive MN markers

Previous RNA sequencing data of little skate MNs (*Jung et al., 2018*), which was analyzed using the zebrafish transcriptome, was re-analyzed with the new little skate genome. Comparing the expression of 10,270 genes in pectoral fin MNs (pec-MNs) with tail-region spinal cord (tail-SC) identified 411 differentially expressed genes (DEGs) including 135 genes upregulated in the pec-MNs and 276 genes in the tail-SC (*Figure 2*). Larger number of DEGs in tail-SC may be caused by comparing heterogeneous cell types in tail-SC with homogeneous cell type in pec-MNs. Although the total number of DEGs are different from the previous data (592 vs 135 genes in pec-MN DEGs), which might be due

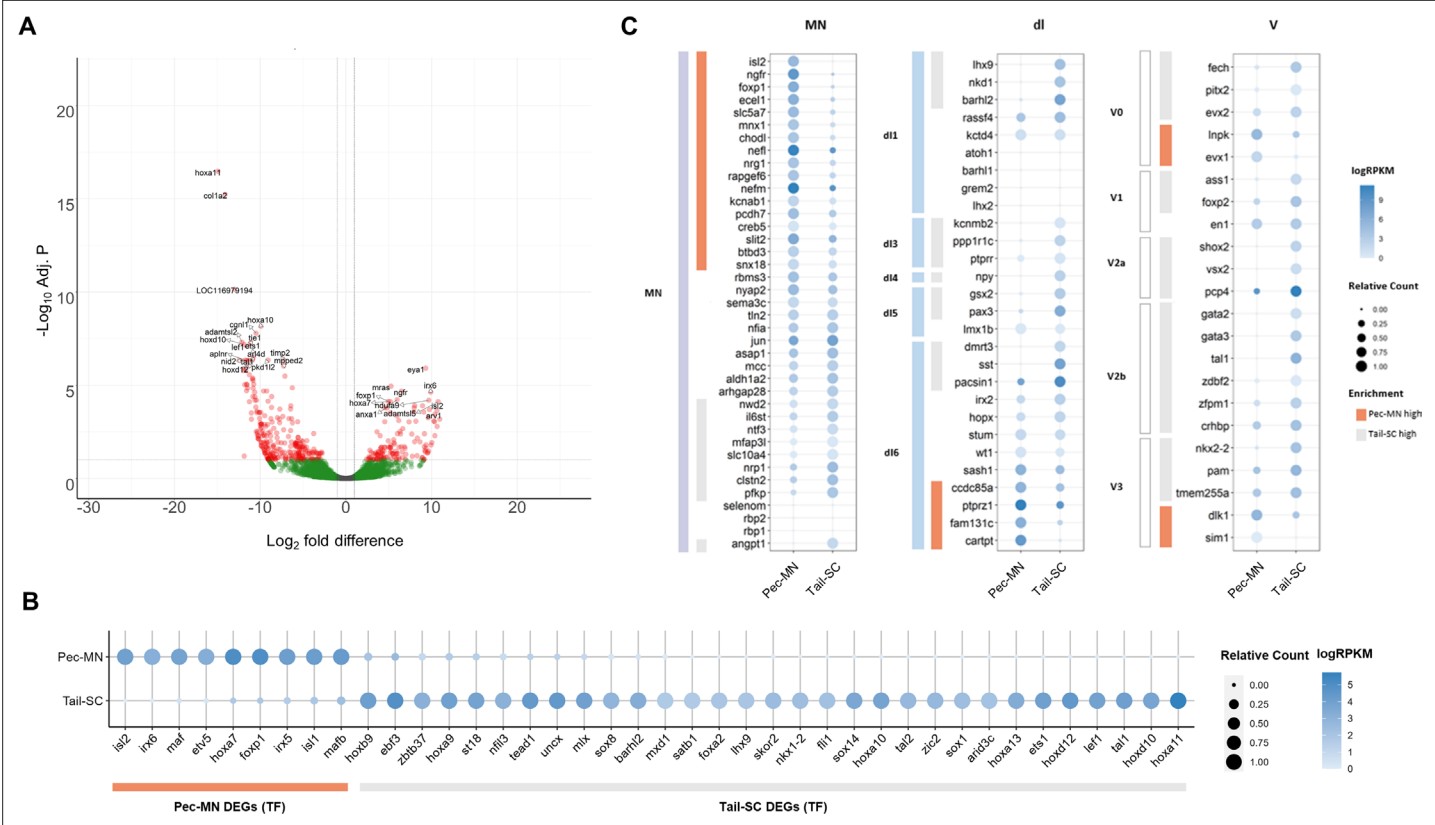

**Figure 2.** Differentially expressed genes of Pec-MNs and Tail-SC. (**A**) Volcano plot of gene expression. Each dot represents individual genes, and each gene is color-coded according to its fold difference and significance: Red dots: FD ≥2, adjusted p-value (adj. p)<0.1, the top 20 genes with lowest p-value are labelled; green dots: FD ≥2, adj. p≥0.1; grey dots: FD <2, adj. p≥0.1. The genes highly expressed in pec-MNs and tail-SC are on the right and left, respectively. (**B**) The differentially expressed TFs under the gene ontology (GO) term 'DNA-binding transcription factor activity'. Pec-MN and tail-SC DEGs are indicated by orange and grey bars, respectively. (**C**) Comparison of pec-MN and tail-SC DEGs with the gene expression of mouse MN and interneuron marker genes (*Delile et al., 2019*); MN: Motor neuron, dl1-6: dorsal interneurons, V0-3: Ventral interneurons. MN and interneuron markers highly expressed in pec-MNs compared to tail-SC are indicated by orange bars; while markers expressed at higher levels in the tail-SC than in pec-MNs (FD ≥2) are indicated by grey bars. Absolute expression levels are color-coded in blue and relative expression levels are indicated by circle size.

to different statistical analysis with different reference genome, previous RNA-seq data based on de novo assembly and zebrafish-based annotation was mostly recapitulated in our DEG analysis based on our new skate genome (21 out of 24 previous fin MN marker genes have the expression level ranked above 70th percentile in Pec-MNs; Supplementary file 3). Among the identified DEGs, genes associated with the GO term 'DNA-binding transcription factor activity' including caudal Hox genes, *Hox10-13* were highly expressed in tail-SC compared to pec-MNs while a rostral Hox gene, *Hoxa7* was highly expressed in pec-MNs (*Figure 2B*), which is consistent with previous immunohistochemical analyses (*Jung et al., 2018*).

Using our newly annotated genome, we comprehensively examined the similarity of gene expression patterns between skate and tetrapod spinal neurons. The expression of a set of molecular markers for spinal MNs and interneurons of mouse embryos (*Delile et al., 2019*) were examined in skate pec-MNs and tail-SC. Overall, we observed a greater number of mouse MN-expressed genes were highly expressed in pec-MNs (enriched in pec-MN, 17 genes; enriched in tail-SC, 9 genes; fold difference (FD) ≥2; *Figure 2C*). On the other hand, a greater number of interneuron markers was observed in tail-SC (enriched in tail-SC, 29 genes; enriched in pec-MNs, 8 genes; FD ≥2), which is composed predominantly of interneuron cell-types.

The evolution of genetic programs in MNs was investigated unbiasedly by comparing highly expressed genes in pec-MNs (percentile expression >70) of little skate with the ones from MNs of mouse and chick, two well-studied tetrapod species. In order to compare gene expression with homologous cell types from each species, we performed RNA sequencing with forelimb MNs of

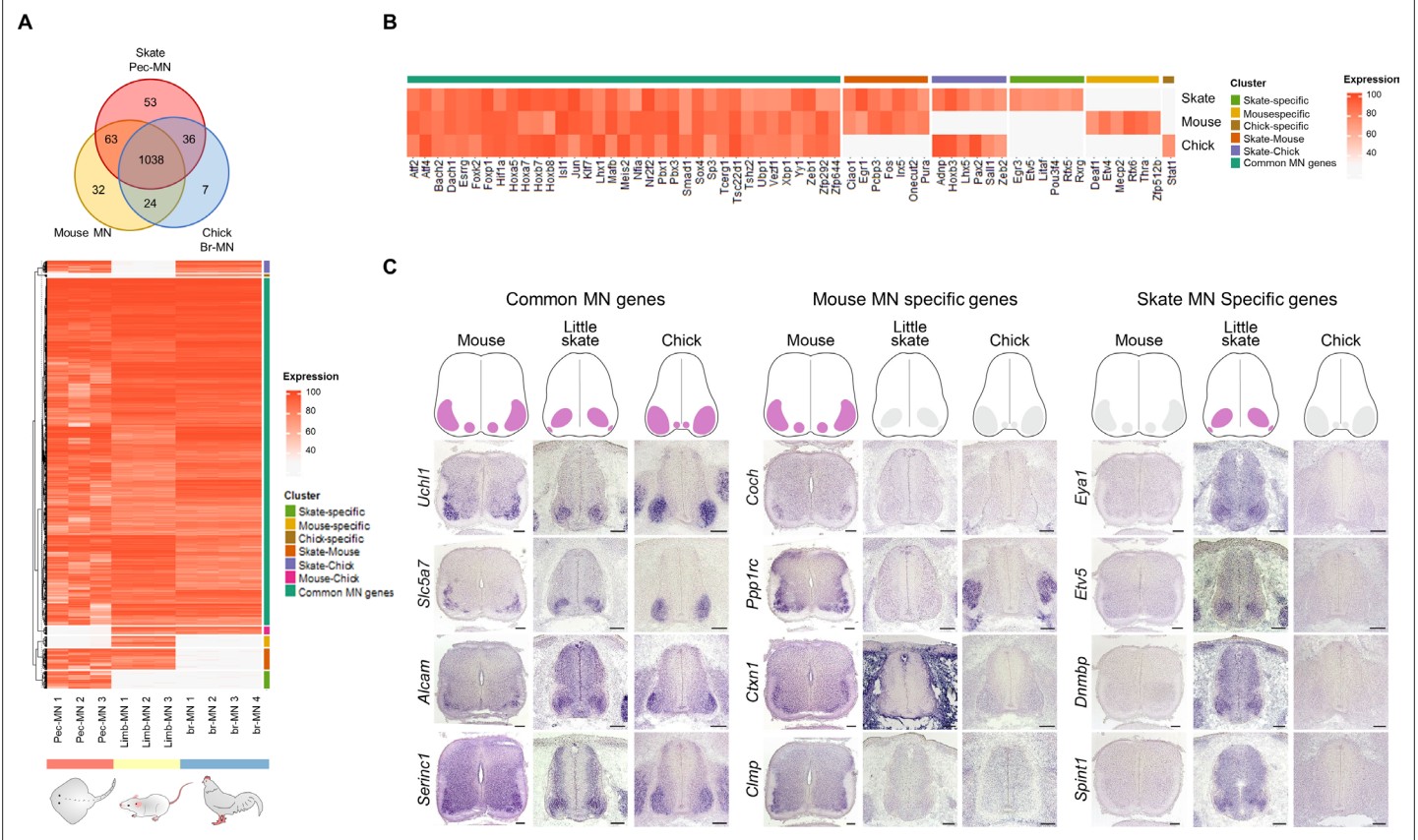

**Figure 3.** Potential MN marker genes commonly or divergently expressed in different species. (**A**) Venn diagram and heatmap of MN-expressed genes (percentile expression >70) of little skate, mouse and chick. In the heatmap, expression levels are color coded according to percentile expression in each species. (**B**) Expression heat map of transcription factor genes. (**C**) Gene expression pattern in tissue sections revealed by in situ RNA hybridization. Shown on the top is the schematic summary of the expression. Common MN-expressed genes: expression levels ranked above 70th percentile in all species; Mouse MN-specific genes: expression levels ranked above 70th percentile in mouse and below 30th percentile in little skate and chick; little skate MN-specific genes: expression levels ranked above 70th percentile in little skate and below 30th percentile in mouse and chick. Tissues are from e13.5 mouse, stage 30 little skate, and HH stage 26–27 chick embryos. Scale bars: 100 µm in (**C**).

The online version of this article includes the following figure supplement(s) for figure 3:

**Figure supplement 1.** Validation of chick MN RNA-seq.

**Figure supplement 2.** Validation of MN gene expression.

**Figure supplement 3.** Common and species-specific genes.

mouse embryos at embryonic day 13.5 (e13.5) and wing level MNs of chick embryos at Hamburger-Hamilton (HH) stage 26–27, which was validated by RNA in situ hybridization (*Figure 3—figure supplement 1*). The comparison of orthologous MN-expressed genes revealed a large number of genes with shared expression as well as species-specific expression which could represent similarity and divergence between skate and the two tetrapod species (*Figure 3*; *Figure 3—figure supplement 3A*). Among shared genes, 1038 genes were commonly found in mouse, little skate and chick, while 63, 36, and 24 genes were identified in the little skate-mouse pair, little skate-chick pair and mouse-chick pair, respectively (*Figure 3A*). Species-specific genes were also identified; 53, 32, and 7 genes were highly expressed (percentile >70) in MNs of little skate, mouse and chick with almost no expression in the remaining species (percentile <30), which suggest that the genes are expressed in MNs of one species but not in MNs of other species at the similar developmental stages (*Figure 3*; *Figure 3—figure supplement 2*). However, 11, 17, and 4 genes of mouse, skate and chick MN-specific genes were found to be paralogs of at least one shared gene (*Figure 3—figure supplement 3B*) leaving only a few species-specific genes without paralog expression. In addition to the comparisons among ortholog genes, we also identified 12, 171, and 6 paralog genes specifically expressed by

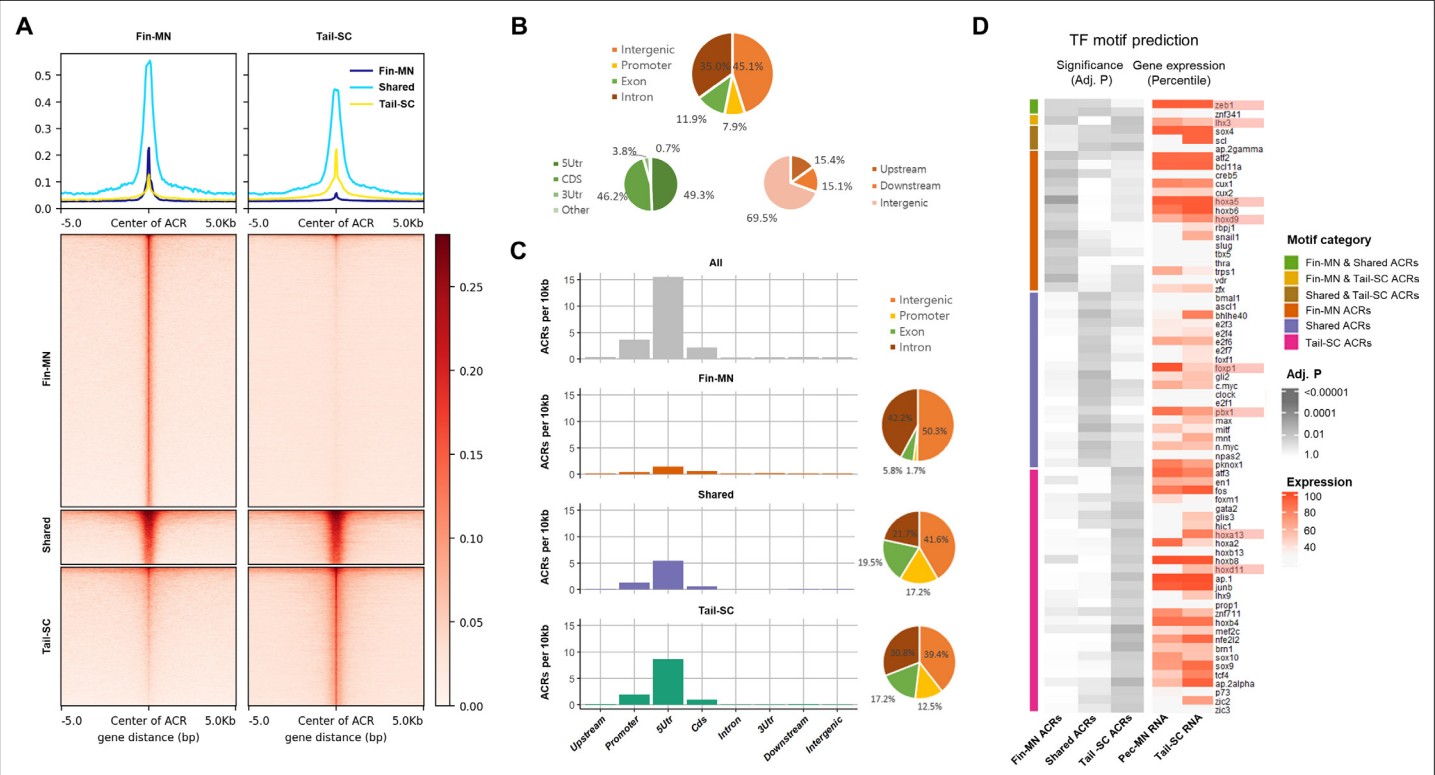

**Figure 4.** Overview of ATAC-seq data. (**A**) Heatmap of 61997 ACRs. Fin-MN ACRs on the left and tail-SC ACRs on the right. The number of ACRs: shared-ACRs, 7869; fin-MNs- specific ACRs, 35741; tail-SC-specific ACRs, 18387. Higher intensity in red shows regions with higher ATAC-seq read depth. The line plot on top shows the average read depth around 5 Kb of ACR centers. (**B**) Proportion of all ACRs (fin-MNs and tail-SC) in different genomic regions. (**C**) The density of ACRs (all, fin-MN-specific, shared, and tail-SC-specific ACRs) in each genomic region normalized by the respective length. (**D**) TF motif prediction in ACRs (adj.p <0.05; around 10 Kb up and downstream of transcription start and end sites of each gene) of pec-MN-expressed genes (percentile >70). From left to right columns: enrichment p-values of fin-MN-specific, shared and tail-SC-specific ACRs. The respective expression in pec-MNs and tail-SC is also shown for each TF on the right two columns.

The online version of this article includes the following figure supplement(s) for figure 4:

**Figure supplement 1.** ATAC-seq data quality control.

**Figure supplement 2.** Chromatin accessibility of constitutively expressed genes and example genes with different peak types.

**Figure supplement 3.** Correlation between promoter accessibility with gene expression.

skate, mouse and chicken MNs, respectively which could further explain species-specific differences of MNs (*Figure 3—figure supplement 3B*; Supplementary files 6 and 7). These results were validated by performing RNA in situ hybridization in tissue sections on a subset of species-specific genes (*Figure 3C*; *Figure 3—figure supplements 1 and 2*).

## ATAC-seq analysis in little skate MNs uncovered putative regulators of MN DEGs

Our high-quality little skate genome assembly, with sufficient contiguity of scaffold covering potential regulatory regions, allowed us to further investigate the mechanisms of gene regulation in little skate fin MNs. We examined chromatin accessibility by performing ATAC-seq from isolated pectoral/pelvic fin MNs (fin-MNs) and tail-SC. The quality control for ATAC-seq data is summarized in *Figure 4—figure supplement 1*; the highest ATAC-seq read depth was observed right before the transcription start site (TSS) for nucleosome-free reads than the nucleosome-bound reads (*Figure 4—figure supplement 1A-D*). The distribution of insert size shows similar trends with the typical size distribution of ATAC-seq analysis (*Yan et al., 2020*; *Figure 4—figure supplement 1E and F*). Through peak calling, a total of 61,997 consensus accessible chromatin regions (ACRs) were identified in fin-MNs and tail-SC, indicated by intense red color in the heatmap (*Figure 4A*). Among them, 7869 ACRs were

found in both fin-MNs and tail-SC while 35,741 and 18,387 ACRs were found only in fin-MNs and tail-SC, respectively (*Figure 4A*; *Figure 4—figure supplement 2*). On average, higher read depth was observed for fin-MN ACRs than that of the tail-SC.

The genome-wide distribution analysis of ACRs showed that many ACRs were found in intron and intergenic regions for all ACR types (intron, 35.0%; intergenic region, 45.1%; *Figure 4B*); however, normalizing each region by its length, the density of ACRs was found to be highest in promoters and 5′UTRs (*Figure 4C*). Even though higher number of fin-MN-specific ACRs were identified, the ACR normalized by region length showed lower density for fin-MN specific ACRs because large number of the ACRs in the fin-MNs were found in intron and intergenic regions which constitute the largest proportion of the genome. After observing that the ACR density is higher near the TSS, the correlation between chromatin accessibility in the promoter region and the gene expression level was investigated. As a result, we found that the genes with greater depth of ATAC-seq in their promoter region are generally expressed at higher levels than those genes with closed chromatin form in both fin-MNs and tail-SC (*Figure 4—figure supplement 3*), indicating that ACRs in the promoter region are likely to be associated with gene activation.

Using the ACR data, enrichment test of putative TF motifs was performed for genes expressed in pec-MNs (percentile >70) to reveal potential regulatory mechanisms (*Figure 4D*). Overall, 19, 25, and 32 TF motif predictions were found in fin-MN-specific ACRs, shared and tail-SC-specific ACRs, respectively. Interestingly, difference in the enrichment of predicted motifs of Hox proteins, well-known MN subtype regulators along the rostro-caudal axis of the spinal cord (*Jung et al., 2018*; *Dasen et al., 2005*; *Dasen et al., 2003*), was found in the ACRs of pec-MNs genes and tail-SC genes. Fin-MN-specific ACRs were enriched with predicted motifs of *Hoxa5* and *Hoxd9*, which are expressed in the fin-MNs of little skate, while the tail-SC-specific ACRs were enriched with the predicted motifs of *Hoxd11* and *Hoxa13*, expressed in tail SCs of little skate (*Jung et al., 2018*). In addition, motif predictions of *Foxp1*, *Pbx1*, and *Lhx3*, well-known regulators in MN (*Dasen et al., 2008*; *Hanley et al., 2016*; *Thaler et al., 2002*), were found in the ACRs; the motif predictions of *Foxp1* and *Pbx1* in shared ACRs and *Lhx3 in* fin-MN-specific and tail-SC-specific ACRs, respectively (*Figure 4D*).

## Predicted shared and diverged gene regulatory systems in MNs of little skate and mouse

To evaluate the degree of conservation of the skate gene regulatory system in MNs with that of land-walking vertebrates, the TF motif prediction was performed with the ACRs of skate fin MN compared with the mouse limb-level MN ATAC-seq data (*Sawai et al., 2022*; *Figure 5*). Among the ACRs of skate pec-MN and mouse limb-level MN-expressed genes (percentile >70; *Figure 5*), a larger number of TF motif predictions were found in mouse (115) compared to skate (*Howe et al., 2013*). However, it is important to note that the larger number of TF motif predictions in mouse MN-expressed genes could be due to the biased motif database toward mouse TFs.

Among the motif predictions, 14 TF motifs were commonly found in both mouse limb-level and skate fin MN ACRs (*Figure 5A and B*), while only a subset of the 14 TF motifs (10 TF motifs, PV interneurons; 8 TF motifs, excitatory neurons) were enriched in ACRs of genes expressed in cortical PV interneuron and excitatory neurons (*Mo et al., 2015*; *Figure 5C*). Motif predictions of *Pknox1*, *Pbx3*, *Hoxa5*, and *Cux1* were found in ACRs of genes expressed in skate fin MNs and mouse limb-level MNs but not in the neurons of the brain, which suggests that the shared TF motifs were not randomly observed, and the shared TF motifs may have functions in regulating cell type or regional identities of neurons.

In the ACRs of the highly expressed genes in MNs (percentile >70), we found a significantly greater number of predicted TF motifs in mouse MNs than in skate MNs (*Figure 5D and E*), suggesting that the greater number of predicted TFs would potentially bind to and regulate the expression of genes expressed in mouse limb-level MNs compared to skate fin MNs.

## Discussion

The little skate is an emerging model to study the evolution of the neuronal circuits for locomotion. Here, in an attempt to understand the origin of the genetic program for tetrapods locomotion, we generated the new reference genome assembly of little skate with gene annotations. We predicted

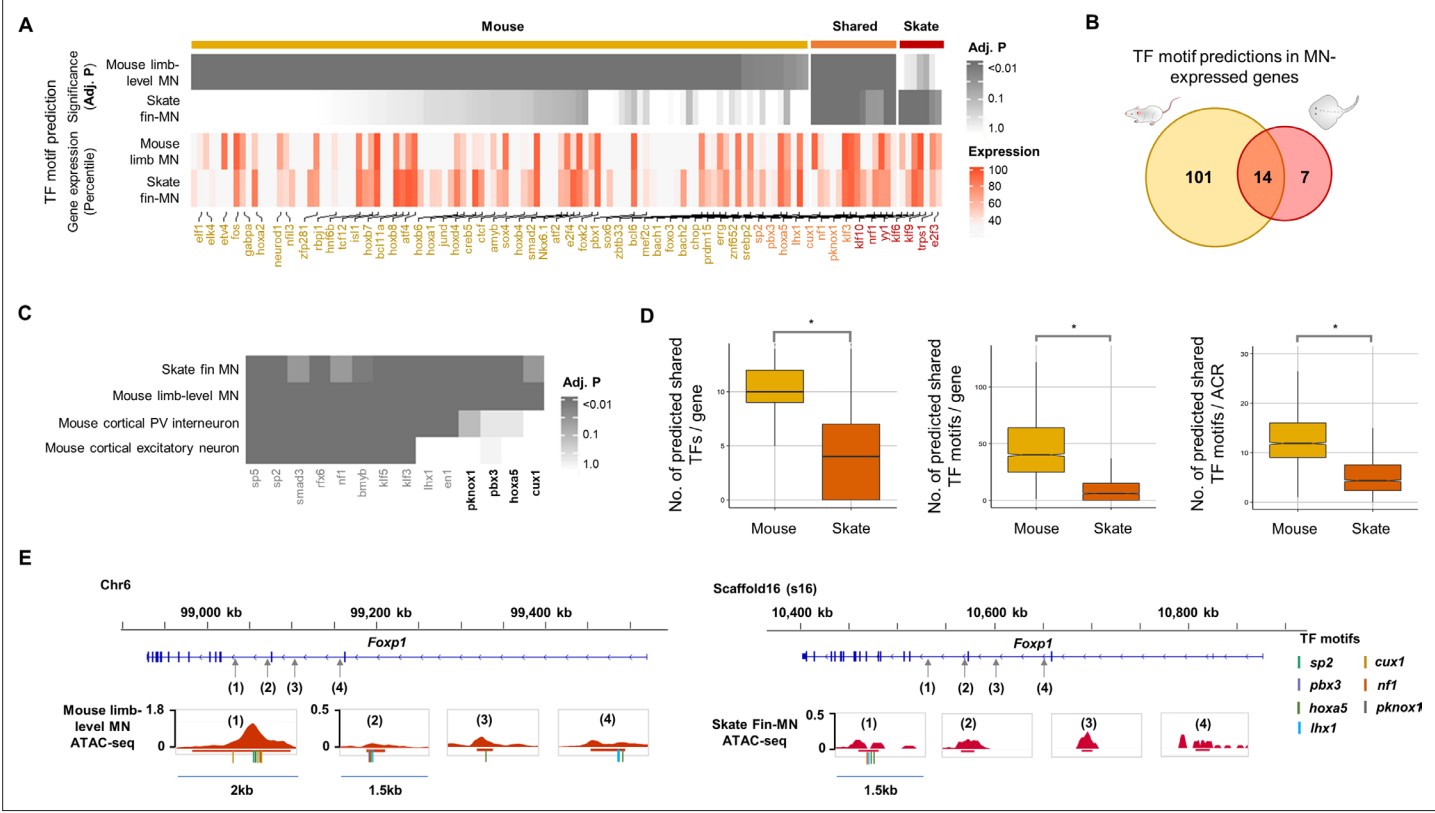

**Figure 5.** Predicted gene regulatory modules in MNs. (**A**) TF motif predictions (adj.p <0.05) identified in skate fin MN and mouse limb-level MN ACRs of MN-expressed genes (percentile >70 in either skate MNs or mouse MNs; around 10 Kb up- and downstream of transcription start and end sites of each gene). The heat map in the bottom shows the expression for the corresponding predicted TF. The significance of the enrichment and expression levels are indicated by the intensity of colors in the heatmap (grey and red). (**B**) Venn diagram of TF motif predictions in the ACRs of MN-expressed genes in mouse and little skate. (**C**) The shared motif predictions compared to motif predictions in ACRs of genes expressed in cortical PV interneuron and cortical excitatory neurons. The heatmap indicates significance of motif enrichment compared to the random background. (**D**) Comparison of the number of shared TFs, the number of shared motif predictions in each gene and the number of motif predictions in each ACR in the genes with percentile expression >70. * significant wilcoxon rank sum test. (**E**) The examples of *TF* motif predictions in *Foxp1*. Shown are the motifs of TFs expressed above 70th percentile in MNs. Predicted motifs for each TF are indicated by colored bars.

The online version of this article includes the following figure supplement(s) for figure 5:

**Figure supplement 1.** Differential regulation of *Foxp1* expression in MNs of different species.

**Figure supplement 2.** Comparison of intergenic size and number of ACRs between mouse and little skate.

the regulatory mechanisms of gene expression in MNs through integrating RNA and chromatin accessibility data from multiple vertebrate species. As a result, we found both conserved and divergent gene expression in MNs across multiple species. Moreover, through chromatin accessibility analysis we found TF motif predictions that could regulate the MN-expressed genes in mouse and little skate. These findings provide deeper insights into the evolution of genetic pathways essential for limb-based locomotion.

## High-quality genome assembly of little skate provides a reference for studying gene regulatory mechanisms in MNs

A previous highly fragmented and incomplete genome of *Leucoraja erinacea* (*Wyffels et al., 2014*) is unsuitable for post-genome analysis. Therefore, improving the quality of the little skate reference genome is a first step towards a more complete analysis of gene regulatory networks. To improve the quality of the little skate genome, whole genome sequencing data including PacBio long reads and Illumina short reads were generated. Even though this study used fewer resources relative to that of more recent sequencing pipelines, which used multiple sequencing technologies including chromosomal conformation capture and physical maps (*Rhie et al., 2021*), this study could assemble

a reference genome which is highly contiguous and offers reliable coding and regulatory region data. This was accomplished by using the assembly of a closely-related species—thorny skate—as a reference during scaffolding process. However, it is important to note that the contig N50 of this little skate genome is still lower than that of the thorny skate assembly. Also, the higher number of scaffolds and the presence of little skate contigs which had not been localized to pseudo-chromosomes of thorny skate may indicate that little skate and thorny skate genomes contain a few structural disagreements.

Despite this limitation, the new reference genome allowed for a more reliable RNA-seq analysis. Through re-analyzing the previously generated RNA-seq data (*Jung et al., 2018*), we found genes which are consistent with the previous immunohistochemical analyses (*Jung et al., 2018*). In addition, we also observed that the expression patterns of MN and interneuron markers in mouse (*Delile et al., 2019*) are highly similar in skate, supporting previous conclusions. Previously unidentified DEGs were found through re-analysis of RNA-seq data using the new reference genome (*Figure 2*), which reflects the limitation of the previous analyses using the zebrafish transcriptome. Aligning the new gene annotation data against the previous de novo transcriptome, and investigating the distribution of alignment length, we also found that the previous de novo transcriptome contains many fragmented genes (*Figure 1—figure supplement 4A*). The previous transcripts generally cover only a small fraction (<20%) of the transcripts predicted in our genome assembly while the new transcripts cover close to 100% of the previous transcriptome. An example case is illustrated in the *Foxp1* gene, a well-known limb/fin MN fate determinant (*Figure 1—figure supplement 4B*). This improved DEG analysis therefore offers a more complete view of gene expression profiles of MNs in little skate.

## The evolution of gene regulatory modules for MN development

During the evolution of motor behaviors, a more complex nervous system likely emerged to control more sophisticated limb movements, such as hand dexterity. Little skate displays walking-like behavior using only around 10 muscles in the pelvic fin, while mammals can use up to 50 muscles in each limb. How the system evolved to generate a more intricate nervous system remains an open question. One mechanism to achieve this might be to increase the number of regulatory modules that control fine-grained gene expression. Recently, an increase in the size of intergenic regions with concomitant increase in the number of ACRs in neuronal genes was proposed as one plausible mechanism to generate a more complex nervous system (*Closser et al., 2022*). Although neuronal genes in little skate have much longer intergenic regions than non-neuronal genes, the intergenic regions in neuronal genes contain a similar fraction of ACRs as in non-neuronal genes as opposed to mouse with significantly increased neuronal gene ACRs (*Figure 5—figure supplement 2*). Other mechanisms seem to be involved in generating complex nervous system. While birds display very complex behaviors, its genome contains short intergenic regions (*Zhang et al., 2014*). In our study, we propose another way of achieving complex nervous system, through more intricate regulatory network. As illustrated in the Figure 5D and a greater number of shared TF motif predictions were found in MN-expressed genes of mouse than in skate allowing an intricate control of gene expression.

*Foxp1*, the major limb/fin MN determinant appears to be differentially regulated in tetrapod and little skate. Although *Foxp1* is expressed in and required for the specification of all limb MNs in tetrapods, *Foxp1* is downregulated in *Pea3*-positive MN pools during maturation in mice (*Dasen et al., 2008*; *Catela et al., 2016*). In addition, preganglionic motor column neurons (PGC MNs) in the thoracic spinal cord of mouse and chick express less than half the level of *Foxp1* expression in limb MNs. Although PGC neurons have not yet been identified in little skate, we tested the expression level of *Foxp1* using a previously characterized tetrapod PGC marker, *pSmad*. We observed that *Foxp1* is not expressed in MNs that express *pSmad* (*Figure 5—figure supplement 1*). Since there is currently no known marker for PGC MNs in little skate, our conclusion should be taken with caution. This predicted complex mechanism of gene expression in mouse may contribute to the emergence of more complex nervous system of mouse compared to that of skate even though the hypothesis remains to be validated in additional organisms.

In summary, the comparative genomic and transcriptomic analysis of the current study suggests evidence of species-specific changes apart from the conservation. Divergent expression of genes in MN (*Figure 3A*; *Figure 3—figure supplement 1*) could allow a greater number of motor pools to be generated in mouse to control complex limb structures. Loss of the *Hoxc* cluster in the little skate genome, correlated with high contents of repeat elements in the genomic locus (*Figure 1—figure*

*supplement 6*), might underly the expanded fin MN domain in the spinal cord without an inter-fin region; while loss of a subset of CTCF binding motifs in the *Hoxa* cluster of little skate (*Figure 1D*) may have induced the relatively expanded domain of *Hoxa9*. The changes in genome structure and sequence could lead to changes in the expression of MN-specific genes as well as Hox genes across the rostro-caudal axis of the spinal cord, affecting the MN organization (*Jung et al., 2010*; *Lacombe et al., 2013*; *Machado et al., 2015*). The changes in the MN organization along the rostro-caudal axis of the spinal cord may affect the connectivity of MNs, which would eventually lead to changes in the pattern of locomotion (*Baek et al., 2017*; *Osseward and Pfaff, 2019*). Continued effort is needed to complete an understanding of the shared and species-specific genetic systems underlying the origin and diversity of neuronal circuits controlling locomotion.

## Limitations

The findings in this study demonstrates the conservation and divergence in gene expression pattern and predicted gene regulatory system among skate and tetrapod species. Whether the predicted TFs actually bind to the putative motifs and whether TFs function as activators or repressors are yet to be validated. As another limitation of the current study, the ACRs analyzed here did not consider long-range regulatory elements. For more accurate characterization of regulatory elements, future studies could provide a more comprehensive approach including ChIP-seq and chromatin conformation capture assay together with functional validation of the motifs.

# Materials and methods

## Animal work

Animal procedures were conducted under the approval of the IACUC at DGIST (DGIST-IACUC-21062403–0002).

## Sampling and DNA extraction and whole genome sequencing

For genomic DNA extraction from little skates (Marine Biological laboratory; 5 stage and sex unmatched skate embryos), the guts were removed and washed 2 times with 1 x PBS. Around 500 μl of G2 buffer (Qiagen) with RNase A (200 μg/ml) was added to the tissue, minced with a razor blade, and transferred to 19 ml G2 buffer in a 50 ml falcon tube. 1 ml of Proteinase K (Qiagen, 19131) was added to the tube and vortexed for 5 seconds, followed by incubation for 2 hr at 50 °C. The samples were vortexed for 10 seconds and loaded onto a genomic DNA column (Qiagen, 10262). Column purification was performed following the manufacturer's protocol. For PacBio long read sequencing, little skate genomic DNA (6.5–9 μg in 150 μl volume) was transferred to a Covaris g-Tube (Covaris, #520079) and sheared in a tabletop centrifuge (Eppendorf, #5424) at 3600 rpm for 30 seconds. After shearing, DNA was cleaned with 0.4 x ratio of Ampure XP magnetic beads (Beckman coulter). Eluted DNA was processed in a PacBio SMRTbell template prep kit v1.0 (PacBio, #100-222-300) following the manufacture's protocol. The SMRTbell templates larger than 15 Kb were enriched using a Blue Pippen size-selection system (Sage Science, #BLF7510). Following size selection, the DNA was cleaned and concentrated using Ampure Xp magnetic beads. The sequencing was performed on the PacBio Sequel system. For Illumina short read sequencing, 1 μg of the high-quality skate DNA was put into Kapa-Roche Library prep kit. The sample underwent 1 PCR cycle. 150 bp paired-end sequencing was performed on the Hiseq4000 with v4 chemistry (Illumina, CA).

## Mouse and chick MN RNA sequencing

For mouse MN RNA sequencing, spinal cords of *Hb9-GFP* mouse (Jackson laboratory) were dissociated and GFP+ MNs were manually collected as described (*Hempel et al., 2007*). The regions for forelimb MNs were visualized under a fluorescence dissection scope and separated from other neurons using a sharp scalpel. The spinal cords were digested with pronase for 30 min at RT. Total RNAs from 400 to 600 MNs were extracted using Picopure RNA isolation kit (Arcturus, KIT0202). The RNAs were amplified using Nugen Trio low input library prep kit (Nugen, 0507). 50 bp paired-end sequencing was performed on the HiSeq4000 with v4 chemistry (Illumina, CA).

For chick MN RNA sequencing, HH stage 12–13 chick embryos were electroporated with a plasmid that drives nuclear YPF expression under the control of *Hb9* promoter (*Lee et al., 2004*). After 3-day

incubation, br-level spinal cords were isolated and dissociated as described (*Hempel et al., 2007*). After electroporation only well electroporated embryos (>50% of the brachial MNs) were selected and used for cell sorting. During the FACS sorting, the YFP positive cells were counted to be around 5–10% of the entire population, shown in *Figure 3—figure supplement 1C*. After the FACS sorting, we confirmed under fluorescence microscope that most of the sorted cells (>90%) were YFP positive. 5000–23,000 YFP$^+$ MNs were collected using a FACS sorter (BD, FACSAria III) and PicoPure RNA Isolation Kit (Arcturus, KIT0202) was used to extract total RNAs. cDNA was synthesized and amplified using Ovation RNA amplification system v2 (Nugen, 3100–12). The sequencing library was prepared using the TruSeq RNA sample prep Kit (Illumina, CA). The suitable fragments (350–450 bp) were selected as templates for PCR amplification using BluePippin 2% agarose gel cassette (Sage Science, MA). 150 bp paired-end sequencing was performed in the NovaSeq6000 (Illumina, CA).

## Immunohistochemistry

For in situ RNA hybridization, 300–1000 bp cDNA region was amplified by PCR with primers listed in Supplementary file 8. The amplified PCR products were used for generating DIG labeled RNA probe by in vitro transcription with T7 RNA polymerase. In situ RNA hybridization and immunofluorescence was performed as previously described (*Jung et al., 2018*). Immunofluorescence images were acquired in a confocal microscope (LSM800, Zeiss).

The antibodies used are as follows:

Primary antibodies: Rabbit anti-Foxp1 (*Dasen et al., 2008*), guinea pig anti-Foxp1 (*Dasen et al., 2008*), mouse anti-Hb9/Mnr2 (81.5C10, DSHB), mouse anti-Lhx3/Lim3 (67.4E12, DSHB), mouse anti-Isl1/2 (39.4D5, DSHB), mouse anti-NFAP (3A10, DSHB), rabbit anti-pSmad (*Dasen et al., 2008*), chicken anti-GFP (GFP-1020, Aves), anti-DIG-AP Fab fragments (#11093274910, Sigma-Aldrich)

Secondary antibodies: Cy3 anti-rabbit antibody (#711-165-152, Jackson immunoResearch), Alexa 647 anti-mouse antibody (#715-605-150, Jackson immunoResearch), Alexa 488 anti-chicken antibody (#703-545-155), Alexa 488 anti-guinea pig antibody (A11073, Invitrogen)

## In ovo chick electroporation

Chick in ovo electroporation was performed as previously described (*Jung et al., 2018*). In brief, *pHb9-nuYFP* (2 µg/µl) plasmid was co-electroporated into the neural tube of HH stage 12–15 chick embryos.

## Collection of public data

The RNA-seq raw data of little skate pec-MNs and tail-SC used in differential gene expression analysis available under accession number PRJNA414974 from the NCBI SRA were downloaded. The ATAC-seq raw data of mouse limb-level MNs (brachial and lumbar) published under the GEO accession of GSE175503 (*Sawai et al., 2022*) was used for comparing with skate ATAC-seq data. The mouse cortical PV interneuron and excitatory neuron RNA-seq and ATAC-seq data (*Mo et al., 2015*) with GEO accession of GSE63137 were downloaded.

The human (GRCh38), mouse (GRCm38), chicken (GRCg6a), zebrafish (GRCz11), and elephant shark (C.milii-6.1.3) reference genomes were collected from Ensemble FTP. The thorny skate (sAmbRad1) reference genome was downloaded from NCBI. Lastly, the three shark genomes were collected from the previous study (*Hara et al., 2018*).

## Sampling of tissues and generation of ATAC-seq data

In stage 30 (*Maxwell et al., 2008*) little skates, pectoral and pelvic MNs were labeled by injecting rhodamine-dextran (3 K, Invitrogen). After 1–2 day incubation at RT, cell dissociation was performed as described (*Hempel et al., 2007*) with some modifications. Spinal cords were isolated and digested with pronase (11459643001, Roche) for 30 min at RT and dissociated MNs from several labeled spinal cords were collected manually and processed for ATAC sequencing library preparation as described (*Corces et al., 2017*). In brief, 150–426 labeled MNs were collected in 25 µl cold ATAC-seq RSB (10 mM Tris-HCl pH7.4, 10 mM NaCl, and 3 mM MgCl$_2$ in Nuclease free water) and centrifuged at 500 rcf for 10 min in a pre-chilled fixed-angle centrifuge. The supernatant was

removed and 10 µl of transposition mix (3.3 µl PBS, 1.15 µl Nuclease free water, 5 µl 2 x TD buffer, 0.25 µl Tn5 (1/10 diluted: 5 µl 2 x TD buffer, 4 µl nuclease free water, 1 µl Tn5), 0.1 µl 1% digitonin) was added and resuspended by pipetting six times. The samples were incubated with shaking at 1000 rpm for 30 min at 37 °C. Tagmented DNA was purified with Qiagen minelute reaction cleanup kit (Qiagen, 28206). PCR mixtures (10 µl Nuclease free water, 2.5 µl 25 µM primer (Ad1), 2.5 µl 25 µM primer (Ad2.X), 25 µl 2 x NEB master mix, transposed sample 20 µl) were prepared and primary PCR was performed in the following condition: 72 °C 5 min, 1 cycle; 98 °C 30 sec, 1 cycle; 98 °C 10 sec, 63 °C 30 sec, 72 °C 1 min 5 cycles; hold at 4 °C. PCR mixture (3.76 µl nuclease free water, 0.5 µl 25 µM primers (Ad1.1), 0.5 µl 25 µM primers (Ad2.X), 0.24 µl 25 x SybrGold in DMSO, 5 µl 2 x NEB master mix, 5 µl pre-Amplified sample) were prepared and quantitative PCR was performed as follows: 98 °C 30 sec, 1 cycle; 98 °C 10 sec, 63 °C 30 sec, 72 °C 1 min 20 cycles; hold at 4 °C. 1/4 max cycles were determined (N) and the primary PCR reaction products underwent N more PCR cycles. For the tail-SC, 240–1000 cells from two embryos were used for ATAC sequencing library preparation. The sequencing was performed on a paired end 50 cycle lane of the Hiseq 4000 using v4 chemistry.

The sequences of primes used in the ATAC library preparation were as follows:

AD1.1:  AATGATACGGCGACCACCGAGATCTACACTAGATCGCTCGTCGGCAGCGTCAGATGTG;

AD2.1: TAAGGCGACAAGCAGAAGACGGCATACGAGATTCGCCTTAGTCTCGTGGGCTCGGAGATGT;

AD2.2: CGTACTAGCAAGCAGAAGACGGCATACGAGATCTAGTACGGTCTCGTGGGCTCGGAGATGT;

AD2.3:  AGGCAGAACAAGCAGAAGACGGCATACGAGATTTCTGCCTGTCTCGTGGGCTCGGAGATGT;

AD2.4:  TCCTGAGCCAAGCAGAAGACGGCATACGAGATGCTCAGGAGTCTCGTGGGCTCGGAGATGT;

AD2.5:  GGACTCCTCAAGCAGAAGACGGCATACGAGATAGGAGTCCGTCTCGTGGGCTCGGAGATGT

## Genome assembly of little skate

The initial contig assembly of little skate genome was performed using raw PacBio reads as input via Flye (v. 2.7.1) (**Kolmogorov et al., 2019**). Default parameter was used for this initial assembly. Haplotypic duplication of the initial contig was identified and removed based on read depth using purge_dups (**Guan et al., 2020**). Scaffolding was performed on the duplicate-removed contigs via reference-guided approach implemented in RaGOO (**Alonge et al., 2019**) using the reference genome of thorny skate or *Amblyraja radiata* available from RefSeq accession GCA_010909765.1. The scaffolded genome of little skate was further polished using Illumina short reads with Free-Bayes (**Garrison and Marth, 2012**). The vertebrate gene data (vertebrata_odb10) of Benchmarking Universal Single-Copy Orthologs (BUSCO v. 5. 2. 2) was used to evaluate the final assembly (**Seppey et al., 2019**). The synteny of BUSCO gene was visualized via ChrOrthLink (**Chul Lee and Rhie, 2021**; https://github.com/chulbioinfo/chrorthlink/) (**Rhie et al., 2021**). The simplified process of genome assembly is summarized in **Figure 1—figure supplement 2**. Comparison among genome assemblies and with previous genome assembly or libraries was made using MashMap (**Jain et al., 2018**).

For gene annotation, comparative annotation approach was used. Cactus (v. 1.0.0) (**Armstrong et al., 2020**) alignment of thorny skate and little skate genome was performed. Comparative annotation toolkit (CAT) (**Fiddes et al., 2018**) was run using the annotation of thorny skate (GCA_010909765.1). Repetitive elements were analyzed using Windowmasker (**Morgulis et al., 2006**). To investigate different classes of repeats, RepeatModeler v2.0.1 (**Flynn et al., 2020**) was used to create species-specific repeat library using default parameters. Repeat annotation was performed by RepeatMasker v4.1.2 (**Smit et al., 2013**). Paralog genes were investigated using eggNOG (**Huerta-Cepas et al., 2019**). Based on the ortholog groups predicted from eggNOG, genes ancestral to Vertebrata were defined as paralogs.

## Identification of ortholog clusters

Using the amino acid sequence of the protein-coding genes of skate, mouse (GRCm38), chicken (GRCg6a), orthologous gene clusters were investigated. OrthoVenn2 was used to identify the orthologous gene clusters using the e-value of 0.01 and the default inflation value (*Xu et al., 2019*). Among the identified ortholog gene clusters, single-copy orthologs identified in all three species were used for next multiple-species RNA-seq analysis.

## RNA-seq data processing and differential gene expression analysis

Quality of RNA-seq raw reads were checked with FastQC (v. 0.11.9). Next, the raw data was trimmed for low quality and Illumina truseq adapter sequences using trimmomatic-0.39 (*Bolger et al., 2014*) with the following options: "ILLUMINACLIP:[AdapterFile]:2:30:10 LEADING:20 TRAILING:20 SLIDINGWINDOW:4:20 MINLEN:50". The resulting clean reads were aligned to the reconstructed little skate genome with STAR (v. 2.7.5 a) (*Dobin et al., 2013*) and were quantified using featureCounts (*Liao et al., 2014*). To explore differential expression between the two tissues (pec-MNs and tail-SC), statistical test was performed via DESeq2 (*Love et al., 2014*) with generalized linear model. Genes with FDR-adjusted p-value of <0.1 were considered significant (*Benjamini and Hochberg, 1995*). For the visualization of RNA-seq data, bigwig files were generated using deeptools (v. 3.4.3) with CPM read depth normalization. For the comparison of known markers, the specific markers provided by *Delile et al., 2019* was used.

For the comparison of gene expression across multiple species, RNA-seq data generated from pec-MNs of little skate, br-MNs of chick and forelimb MNs of mouse were used. The comparison was made using 9253 orthologous genes present in all species. The orthologous genes were further filtered to obtain the list of genes showing minimum expression level of 20 average read counts for each species. To account for different gene length in multiple species, RPKM normalization was used. Genes with the average percentile expression of above 70 were considered highly expressed in MNs. The common genes were defined as those genes with percentile expression greater than 70 in all three species. Species-specific genes were defined as those genes with average percentile expression >70 at the same time, the average percentile expression of the remaining species below 30. The visualization of the gene expression was done via R 'ComplexHeatmap' package. Similarity index between two species shown in *Figure 3—figure supplement 3A* was calculated using the following formula:

$$SI = 1 - \sqrt{\left(1 - \frac{b}{a+b}\right) * \left(1 - \frac{b}{b+c}\right)}$$

SI represents the similarity index. The number of species-specific genes of two species are indicated by a and c, while the number of common genes is represented by b.

## ATAC-seq data processing and differential accessibility analysis

For chromatin accessibility assay, ATAC-seq data was used. Similar to RNA-seq data, the quality of the raw sequence data was investigated with FastQC (v. 0.11.9) followed by trimming of sequences with low quality and NexteraPE adapter sequences using trimmomatic-0.39 (*Bolger et al., 2014*) with following options: 'ILLUMINACLIP:[AdapterFile]:2:30:10 LEADING:20 TRAILING:20 SLIDINGWINDOW:4:20 MINLEN:30'. The clean reads were aligned to the reference genome of little skate with BWA (v. 0.7.17). The quality of the ATAC-seq reads were further checked using R 'ATACseqQC' package. After the alignment, duplicated reads were removed with Picard (v. 2.23.2) (RRID:SCR_006525, v. 2.23.2; http://broadinstitute.github.io/picard/). Moreover, reads with low mapping quality and those reads that were originated from MT genome were removed with samtools (*Li et al., 2009*) (v. 1.9). Peak calling was performed for each tissue after pooling the replicates of respective tissue (fin-MNs and tail-SC) using MACS2 (v. 2.2.7.1) with the following options: '–keep-dup all –nomodel, -q 0.05– -f BAMPE -g 1,974,810,099'. ACRs were defined by merging the peaks from the two tissues using bedtools merge (v. 2.29.2). For the visualization of ATAC-seq data, bed files of ACRs and bigwig files generated by deeptools (v. 3.4.3) were used.

The ACRs identified were further classified into different regions based on gene annotation (including upstream, promoter, 5'UTR, CDS, intron, 3'UTR, downstream and intergenic). The region covering 1000 bp upstream of start of a gene was defined as promoter region and 10,000 bp upstream

of promoter region or downstream of 3' end of a gene was defined as upstream or downstream, respectively. ACRs located in the remaining non-genic regions were considered intergenic ACRs.

## TF binding motif predictions

For the ACRs of interest and promoter regions of the candidate genes, motif enrichment analysis was performed with HOMER and 'annotatePeaks.pl' script was used to locate the binding motifs (*Heinz et al., 2010*) using the collection of homer vertebrate TFs and the binding motifs of Hox gene families available in HOCOMOCO database v11 (*Kulakovskiy et al., 2018*). The putative CTCF motif prediction was also performed using HOMER vertebrate CTCF binding motifs.

## Code availability

This study does not involve previously unreported computer code or algorithms. The list of software, their versions and options used are described in the Materials and methods section.

## Acknowledgements

We thank NYU Langone's Genome Technology Center for technical support. This study was supported by the DGIST R&D Program of the Ministry of Science and ICT (2021010004) to MB. It was also supported by Basic Science Research Program through the National Research Foundation of Korea (NRF) funded by the Ministry of Education (NRF-2019R1I1A2A01041345) to MB, the Marine Biotechnology Program of the Korea Institute of Marine Science and Technology Promotion (KIMST) funded by the Ministry of Ocean and Fisheries (MOF) (No. 20180430), Republic of Korea to HK, and National Institutes of Health (NIH) grant R35 NS116858 to JD. The PacBio Sequel data was obtained thanks to the National Institutes of Health Shared Instrumentation Grant 1S10OD023423-01.

## Additional information

### Competing interests

Heebal Kim: The other authors declare that no competing interests exist.

### Funding

| Funder | Grant reference number | Author |
| --- | --- | --- |
| Ministry of Science and ICT, South Korea | 2021010004 | Myungin Baek |
| National Research Foundation of Korea | NRF-2019R1I1A2A01041345 | Myungin Baek |
| Ministry of Oceans and Fisheries | 20180430 | Heebal Kim |
| National Institutes of Health | 1S10OD023423-01 | Adriana Heguy |
| National Institutes of Health | R35 NS116858 | Jeremy S Dasen |

The funders had no role in study design, data collection and interpretation, or the decision to submit the work for publication.

### Author contributions

DongAhn Yoo, Conceptualization, Software, Formal analysis, Visualization, Writing – original draft, Writing – review and editing; Junhee Park, Conceptualization, Resources, Writing – review and editing; Chul Lee, Software, Formal analysis, Writing – review and editing; Injun Song, Tery Yun, Hyemin Lee, Validation; Young Ho Lee, Adriana Heguy, Resources, Writing – review and editing; Jae Yong Han, Writing – review and editing; Jeremy S Dasen, Conceptualization, Resources, Supervision, Writing – original draft, Writing – review and editing; Heebal Kim, Conceptualization, Supervision, Funding acquisition, Writing – original draft, Writing – review and editing; Myungin Baek, Conceptualization,

Resources, Data curation, Formal analysis, Supervision, Funding acquisition, Writing – original draft, Writing – review and editing

### Author ORCIDs
Jae Yong Han http://orcid.org/0000-0003-3413-3277
Jeremy S Dasen http://orcid.org/0000-0002-9434-874X
Heebal Kim http://orcid.org/0000-0003-3064-1303
Myungin Baek http://orcid.org/0000-0001-8906-901X

### Ethics

Animal procedures were conducted under the approval of the IACUC at DGIST (DGIST-IACUC-21062403-0002).

### Decision letter and Author response

Decision letter https://doi.org/10.7554/eLife.78345.sa1
Author response https://doi.org/10.7554/eLife.78345.sa2

## Additional files

### Supplementary files

• Supplementary file 1. Ortholog gene cluster. Ortholog genes defined by the OrthoVenn2.

• Supplementary file 2. DEG results. The DEG results for little skate (pec-MN vs tail-SC) and chick (br-MN vs ce-SC) MNs.

• Supplementary file 3. Comparison with previous RNA-seq data. The percentile expression level and p-values of previous MN marker genes.

• Supplementary file 4. Multiple species RNA. The percentile expression of genes displayed in *Figure 3*.

• Supplementary file 5. eggNOG ortholog groups. The ortholog group information predicted by eggNOG.

• Supplementary file 6. Species-specific Paralogs. The list of additional paralog genes which are species-specific.

• Supplementary file 7. Specific genes without common gene paralogs. The list of species-specific genes which do not share the same ortholog groups with common genes.

• Supplementary file 8. Oligonucleotide sequences used in generating in situ probes. T7 promoter sequence is indicated in blue

• Transparent reporting form

### Data availability

The genome sequence of little skate is archived on NCBI BioProject under accession PRJNA747829. The RNA-seq data of chick br-MN and mouse limb MNs, and ATAC-seq data of little skate are available from the NCBI Gene Expression Omnibus database under accession number GSE180337.

The following datasets were generated:

| Author(s) | Year | Dataset title | Dataset URL | Database and Identifier |
| --- | --- | --- | --- | --- |
| Yoo D, Park J, Lee C, Song I, Lee YH, Yun T, Yun H, Heguy A, Han JY, Dasen JS, Kim H, Baek M | 2022 | Leucoraja erinacea Genome sequencing and assembly | https://www.ncbi.nlm.nih.gov/bioproject/?term=PRJNA747829 | NCBI BioProject, PRJNA747829 |
| Yoo D, Park J, Lee C, Song I, Lee YH, Yun T, Lee H, Heguy A, Han JY, Dasen JS, Kim H, Baek M | 2022 | Little skate motor neurons | https://www.ncbi.nlm.nih.gov/gds/?term=GSE180337 | NCBI BioProject, GSE180337 |

The following previously published datasets were used:

| Author(s) | Year | Dataset title | Dataset URL | Database and Identifier |
|---|---|---|---|---|
| Jung H | 2018 | Genomewide screening for pectoral fin MN specific markers | https://www.ncbi.nlm.nih.gov/bioproject/PRJNA414974 | NCBI BioProject, PRJNA414974 |
| Closser M | 2020 | An expansion of genomic regulatory complexity underlies vertebrate neuronal diversity (house mouse) | https://www.ncbi.nlm.nih.gov/bioproject/PRJNA630707/ | NCBI BioProject, PRJNA630707 |
| Genome Reference Consortium | 2020 | Human assembly and gene annotation | ftp://ftp.ensembl.org/pub/release-102/fasta/homo_sapiens/dna/ | Ensembl release 102, GRCh38.p13 |
| Genome Reference Consortium | 2020 | Mouse assembly and gene annotation | ftp://ftp.ensembl.org/pub/release-102/fasta/mus_musculus/dna/ | Ensembl release 102, GRCm38.p6 |
| Genome Reference Consortium | 2020 | Chicken assembly and gene annotation | ftp://ftp.ensembl.org/pub/release-102/fasta/gallus_gallus/dna/ | Ensembl release 102, GRCg6a |
| Genome Reference Consortium | 2020 | Zebrafish assembly and gene annotation | ftp://ftp.ensembl.org/pub/release-102/fasta/danio_rerio/dna/ | Ensembl release 102, GRCz11 |
| Institute of Molecular and Cell Biology | 2020 | Elephant shark assembly and gene annotation | ftp://ftp.ensembl.org/pub/release-102/fasta/callorhinchus_milii/dna/ | Ensembl release 102, Callorhinchus_milii-6.1.3 |
| Vertebrate Genomes Project | 2019 | Amblyraja radiata (thorny skate) genome, sAmbRad1 | https://www.ncbi.nlm.nih.gov/bioproject/PRJNA593039/ | NCBI BioProject, PRJNA593039 |
| Hara Y | 2018 | Shark genomes provide insights into elasmobranch evolution and the origin of vertebrates | https://ddbj.nig.ac.jp/resource/sra-submission/DRA006338 | Data Bank of Japan, PRJDB6260 |
| Sawai A | 2022 | PRC1 Sustains the Memory of Neuronal Fate Independent of PRC2 Function (house mouse) | https://www.ncbi.nlm.nih.gov/bioproject/PRJNA732648 | NCBI Bioproject, PRJNA732648 |

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
