## [Editor Report]

This study provides the genome of the little skate *Leucoraja erinacea*, a cartilaginous fish that displays pelvic fin-driven walking-like behavior. Leveraging this genomic resource, the authors compare gene expression and chromatin accessibility profiles in motor neurons of the little skate and other species (e.g., mouse, chicken), aiming to predict conserved and divergent gene regulatory mechanisms underlying motor neuron development. The work represents an important contribution to the field of comparative genomics and evolutionary biology.

---

## [Decision Letter]

**Decision letter after peer review:**

Thank you for submitting your article "Little skate genome exposes the gene regulatory mechanisms underlying the evolution of vertebrate locomotion" for consideration by *eLife*. Your article has been reviewed by 3 peer reviewers, and the evaluation has been overseen by a Reviewing Editor and Catherine Dulac as the Senior Editor.

The reviewers have discussed their reviews with one another, and the Reviewing Editor has drafted this to help you plan the next steps. The reviewers agree that the paper has potential, but extensive revision and new data are needed.

Essential revisions:

1. The authors report the identification of conserved and divergent molecular markers across multiple species with RNA-Seq, but they do not validate the expression of novel markers in either category with an independent method (e.g. in situ or antibody staining).

Such validation is needed to substantiate the authors' conclusions.

2. The reviewers do recognize that functional analyses are not feasible in the little skate. However, to substantiate the claim that more complex regulatory mechanisms have evolved in tetrapods to accommodate sophisticated motor behaviors, the authors should conduct additional gene expression experiments. For example, as pointed out by reviewer 1, the authors can focus on Foxp1 and Snail.

3. Reviewers 1 and 3 raised a number of important issues regarding the bioinformatic comparisons (e.g., completeness of datasets, quality control, introduction of bias) that must be carefully addressed. First, the authors must show validation of their RNA-Seq method in chick, as suggested by reviewer 1. Second, the reviewers would like to see a re-analysis of the RNA-Seq datasets in an unbiased manner, for example, by directly comparing MN expression of orthologous genes in the different species (e.g., compare highly expressed genes in MNs of the different species). Two reviewers stressed that comparing a pure MN population to tail tissue in the skate and DRGs in the mouse was not an appropriate comparison. Third, comparisons of mouse and skate-accessible chromatin regions should be done without biasing first to DEX genes. Lastly, additional key aspects of evolution, such as paralogue substitution or expression of species-specific genes should also be considered. Authors could follow comparisons similar to: 10.1038/s41559-021-01580-3.

4. The reviewers agree that a significantly improved version of the little skate genome is an important contribution. In the first part of the manuscript, the authors should mention the work by Marletaz et al. (bioRxiv) that also provides a new little skate sequenced genome.

5. There is a propensity to overstate claims throughout the manuscript. For example, without functional data in little skate, the claim that the TF networks are much simpler is not substantiated simply by ATACseq and predicted binding sites. The authors should tone down statements in Abstract and Discussion (as indicated by the reviewers) and always make it very clear that they are simply predicting regulatory connections between all these genes.

*Reviewer #1 (Recommendations for the authors):*

1) Based on the low convergence of mouse, skate and chick MN transcriptomes it may be a possibility that one of the datasets is incomplete. The authors should go back to their RNAseq datasets and check for the expression of known MN markers that are conserved in the 3 species to identify where this inconsistency arises. It would also be helpful to show validation of their chick RNAseq method, for example, show the efficiency and specificity of GFP expression in MNs, and what % of MNs were electroporated and sorted before sequencing.

2) Another surprising finding is that a higher number of putative TF binding sites are found in the tail-SC-specific ACRs for pectoral MNs and that the expression levels of these TFs do not correlate with this trend. The authors should comment on this discrepancy. Similarly, while they were found to be highly expressed in pectoral MNs, 5 genes including Isl1, show no associated ACR (Figure 5C). Do the authors think that this is due to long-range regulation of those genes? Some discussion of this would be clarifying.

*Reviewer #2 (Recommendations for the authors):*

As already stated, the main limitation of the study seems to be the strategy followed for the RNAseq and ATACseq comparative analysis. The use of a prior intra-species comparison with heterogeneous criteria is not well-justified, it is a potential source of bias and should be avoided. Why authors do not directly compare MN expression and ATACseq of orthologous genes in the different species? In addition, it is known that there are extensive paralogue substitutions that indicate common regulatory mechanisms but might be missed if not analysed directly. Authors could follow comparisons similar to previously published articles: 10.1038/s41559-021-01580-3.

Comments on each section:

1) Abstract:

Several statements are not strongly supported by data:

"Conserved MN genes were enriched for early-stage nervous system development."

"conservation of the potential regulators with divergent transcription factor (TF) networks through which expression of MN genes is differentially regulated."

"TF networks in little skate MNs are much simpler than those in mouse MNs."

2) Figure 1 data:

In the first part of the manuscript, the authors should mention biorxiv Marletaz et al.'s work which also provides a new little skate sequenced genome and analysis, with a higher number of identified coding genes compared to this work and also with 3D chromatin structure data that identifies TAD boundaries that coincide with CTCF sites in Hoxa and Hoxd clusters mentioned in Baek et al.

3) Figure 2 data:

(3.1) The comparison of Pectoral MN with tail SC, compares a purified population with a mix of cell types including MN. This unbalanced comparison could explain why tail SC shows 3 times more DEX genes than pec-MN.

(3.2) In the previous analysis published by the group, 592 genes were identified as 2 fold enriched in pectoral MN, why the new analysis retrieves only 135 genes? This should be clarified for the reader.

(3.3) Also from previous work of the authors several genes were determined to be expressed in little skate MN (HB9, lhx3, lhx1, Ephs, Ephrins, slit3, onecut1, nell2, nrcam, ngfr, pappa2, cdh7, amigo1, unc5c, lrx5, lrp1b, Zfp804a, etc), which of these genes are predicted to be enriched (or expressed at least) in the Pec-MN data set? Were all the genes found? This could be used as quality control of the data set or data set analysis.

4) Figure 3 data:

(4.1) Compares "Highly expressed genes in pec-MN". What are the criteria followed to assigned "highly expressed genes"?

(4.2) This data set is now compared with DEX of mouse embryonic MN compared to sensory neurons. Which type of mouse MN? at what A-P level?

(4.3) If skate genes are selected by high expression, why do mouse MN not follow the same criteria to avoid bias of comparison? As a control for bias in the criteria: What is the overlap of mouse highly expressed genes (with similar criteria to skate gene selection) with the data set of mouse MN DEX compared to sensory mouse neuron?

(4.4) Same for chick data, how equivalent are brachial MN to mouse MN dataset? Why select DEX compared to the brachial spinal cord and not sensory neurons as in mouse.

Without any justification for the followed strategy, it seems different populations and comparisons will introduce important biases that limit the conclusions.

The best strategy would be to compare highly expressed genes in MN in the different species and clarify if used populations are really "homologous" cell types in the different species.

(4.5) The variability of sampled populations or the biases introduced in the analysis might explain the low overlap found among the 3 species (Figure 3A). For the Venn diagram, is the overlap among categories higher/lower/equal to what would be expected by chance?

5) Figure 4 data:

(5.1) Similar to previous comments: Why limit TF motif enrichment of ACRs to only DEXs in pec-MN and tail-SC that (1) highly reduced the number of ACRs and (2) Induces a bias in the analysis? Analysis should be performed with all MN ACRs.

(5.2) Line 224: On the other hand, the tail-SC-specific ACRs in the pec-MN DEGs were enriched with the binding sites of Hoxa9, Hoxa11, Hoxd10 and Hoxd11, most of which are expressed in fin-MNs of little skate.

Could the authors provide a sentence to try to explain this observation? I found confusing that this is mentioned without providing more context.

6) Figure 5 data:

(6.1) Similar to previous comments, conclusions driven by this analysis might be biased by wrong comparative strategies to start with, thus conclusions are not well supported. Comparative of mouse and skate ACRs should be done without biasing first to DEX genes, which highly reduced the number of analysed ACRs, particularly to shared genes that are only 40.

(6.2) MN TF enrich motifs are broadly found in mouse sensory and skate tail DEX genes, which might be a strong indication that the strategy used in the analysis is not suitable for the identification of MN gene regulatory networks.

(6.3) Figure 5C: Most TF binding motifs in databases are built with experimental data from mouse or human TFs. Although core sequences for TF binding sites in the same families are conserved, small differences arise in individual members or between species. Could this explain why there is a higher number of motifs found in mouse than in skate? In addition, is the number of motifs in mouse and skate normalized by the size of analysed sequence?

7) General comments on format:

(7.1) Figures 3-5: differences among circles sizes are difficult to appreciate, maybe color heatmaps would be more informative.

(7.2) Figure 4 B, C could be included in supplementary.

*Reviewer #3 (Recommendations for the authors):*

The authors should be careful in their phrasing so as to always make it clear that they are simply predicting regulatory connections between all these genes.

For instance, on Page 3, line 29:

"Comparison of accessible chromatin regions between mouse and skate MNs revealed conservation of the potential regulators with divergent transcription factor (TF) networks through which expression of MN genes is differentially regulated. TF networks in little skate MNs are much simpler than those in mouse MNs, suggesting a more fine-grained control of gene expression operates in mouse MNs."

As one can see, without functional data in little skate, the claim that the TF networks are much simpler is not substantiated simply by ATACseq and predicted binding sites.

A similar claim on Page 17, line 338 indicates the same propensity to overstate these claims:

"As illustrated in the Figure 5C, a greater number of shared TFs binds to their downstream TFs in mouse MNs than in skate pec-MNs, allowing an intricate control of gene expression and thus the more complex nervous system of mouse compared to that of skate"

Again, there was no demonstration of little skate TFs binding to any downstream TF genes, only predicted binding sites. Thus, the claim is unsubstantiated.

These examples indicate an overall propensity by the authors to gloss over the fact that no functional connections between any TFs or genes were experimentally demonstrated in little skate. I do not believe it is appropriate to substitute binding motif enrichment for experimentally determined regulatory connections when inferring gene regulatory networks in something as complex as vertebrate development. There are a number of reasons why this is problematic.

[Editors’ note: further revisions were suggested prior to acceptance, as described below.]

Thank you for resubmitting your work entitled "Little skate genome exposes the gene regulatory mechanisms underlying the evolution of vertebrate locomotion" for further consideration by *eLife*. Your revised article has been evaluated by Catherine Dulac (Senior Editor), myself (reviewing editor), and 3 reviewers.

The revised manuscript has been significantly improved by the validation of RNA-Seq findings, the re-analysis of RNA-Seq and ATAC-Seq datasets, and the requested text changes. However, there are some remaining issues that need to be addressed, as outlined below (please see detailed comments by reviewers #1 and #3):

Essential revisions:

1. Please clarify whether you performed ATAC-seq from mouse MNs.

2. The findings on Snail1 are not described, and this part should be removed.

3. Please address the concerns raised by reviewer #3 about Figure 5.

4. We acknowledge that the revised version has toned down most claims on gene regulatory mechanisms, but we further suggest toning down the claim made in the title of the paper.

5. The manuscript should be edited for language and grammatical errors and carefully proofread.

Because your manuscript provides valuable datasets (e.g., little skate genome, new RNA-Seq data, new ATAC-Seq data), myself and all reviewers agree that your manuscript should be considered under the "Tools and Resources" article type.

*Reviewer #1 (Recommendations for the authors):*

The manuscript is much improved, as the authors have addressed a number of the major previous concerns, including re-analyzing RNA-seq data, generating a new set of mouse RNA-seq data, and validating their results with in situ hybridization. The new analysis seems much more convincing than the previous one. Some lingering concerns remain, mostly regarding figure 5 and the central claim of the paper, which is still not functionally supported. The authors did however tone down their claims and pointed out the limitations of the study.

(1) Did the authors perform ATAC-seq from mouse MNs? They show some data in figure 5 but it was not clear from the description of the methods or the results whether they performed the experiment.

(2) It seems that repeating the motif analysis of the ATAC-seq data gave completely new regulatory factors for FoxP1, even though the same regions as in the original manuscript are shown. It appears that Snai1 is no longer on the list so the experiment in the supplemental figures testing its function is somewhat moot. The authors also do not describe the experiment at all in the Results section so it should be taken out.

*Reviewer #3 (Recommendations for the authors):*

The revised manuscript has made important efforts in answering some of the raised concerns but did not address other key issues, particularly at the end of the manuscript.

The description of the little skate genome is, with no doubt, useful for the field of Evolutionary Biology. Then authors, re-analyse previously published experimental data and compare differential gene expression between isolated pectoral motorneurons and a mixed population of the tail spinal cord of little skate (Figure 2) and differential accessible chromatin regions between Fin motorneurons (Pectoral+pelvic) compared to a mixed population of the tail spinal cord (Figure 4). These two figures are descriptive and it is unclear what insights are provided by the comparison of these heterogeneous populations of cells. Thus do not seem to constitute a great advancement in the understanding of gene regulatory networks present in the little skate or their evolution.

In the new version of the manuscript authors now compared homologous motorneuron populations of the little skate, mouse, and chick, which reveals a remarkable degree of conservation of the transcriptomes in the 3 species (Figure 3), which is even higher when considering possible paralog substitutions in the few divergently expressed genes. This is radically different from the initial version of the manuscript and I think is an important observation.

Finally, the authors attempt to describe and compare motorneuron gene regulatory networks in mouse and skate in Figure 5. I think this is the weakest part of the manuscript, not only I found it hard to follow, not clearly explained, and using vague terms and overstatements but in addition the methodology and strategy for the analysis seem incorrect, leading to potentially wrong conclusions.

To start, it is unclear to me what is exactly the mouse motorneuron population that is being compared and how equivalent it is to the mixed Pectoral and pelvic motorneuron population from the little skate.

More importantly, Figure 5A: authors describe 18 TF motifs commonly enriched in mouse and skate MN ACRs, among which 6 TFs predicted to bind those motifs are expressed both in mouse and skate. What does this mean? Is this overlap biologically meaningful? How many would overlap if enriched motifs in skate ACRs are compared to motifs in mouse ACRs of a non-related neuron type at a similar developmental stage, such as cortical interneurons cortical projection neurons? Less? More? similar number?

Authors then look for motif enrichment of these 6 common motif/TFs in ACRs of other expressed TFs (either expressed both in mouse and skate or only in one of them). It is unclear why TFs with enriched motifs in mouse and skate should be upstream of the other TFs, it is equally possible that the 6 common motif/TFs are downstream of the expressed TFs. In addition, using the term "interactions" for motif enrichment of one TF in the ACRs of another TF is an overstatement.

It is also striking that "interactions" of the 6 mouse common motif/TFs seem equally prevalent within ACRs of mouse-expressed TFs compared to ACRs of TFs expressed in skate but not in mouse. Similarly, for the skate data, motif enrichment in ACRs of commonly expressed TFs or TFs only expressed in the skate is not higher than in ACRs of TF expressed in mouse only. These results again raise doubts if this analysis is meaningful at all. Would it be different using just randomly picked ACRs from other TFs not expressed in MN? Finally, as already mentioned in the first review, the increasing number of motifs for mouse TFs could be again due to methodological limitations (based on known motifs in mouse compared to skate, etc) but not biologically meaningful, it should be stated right away when described and not as a separate paragraph in the section of limitations for the study.

Considering all these methodologic concerns, together with the lack of any experimental validation of predicted binding sites for any of the TFs, it would be advisable to completely remove that figure.

In summary, the manuscript has valuable information, namely the description of the little skate genome and the comparison of mouse, chick, and little skate MN transcriptome. It also provides a description and comparison of some heterogeneous populations of neurons in little skate which provide limited insights into gene regulation in skate motorneurons. Most importantly, I think the last part of the manuscript (Figure 5) is below the standards required for a journal such *eLife*. Unfortunately, I don´t think the paper, in its current revised format, provides any mechanism underlying the evolution of vertebrate locomotion as stated in the title.

Other comments:

Authors constantly use the term "MN genes" or similar expressions when should be referring to "MN expressed genes" or "MN enriched genes". Other non-accurate expressions:

"Number of predicted TFs enriched" when it refers to TF motif predictions, "expression of each predicted TF" when meaning "TF expression for the corresponding TF enriched motif ", "predicted shared TFs found in MN ACRs" meaning " predicted shared enriched TF motifs found in MN ACRs", etc.

This lack of accuracy makes it harder for the reader to follow the text.

Methods include snail cDNA electroporation, but I don't think those results are included in the manuscript.

In figure 4D, the legend states ACR group but would be clearer to label as Motif category or Motif distribution in ACR categories.

Line 251: "Together with the predicted binding sites of the Hox proteins, the binding sites of Foxp1 and Pbx1 and Lhx3, which are well-known regulators in MN(31-33), were enriched in shared ACRs and fin-MN-specific and tail-SC-specific ACRs, respectively (Figure 4D)"

However in the figure Pbx1 is also in the shared category as Foxp1, not in fin-MN specific, correct?

Why Figure 5 shows common TFs only 16 TFs, if figure 3 shows many more?

---

## [Author Response]

Essential revisions:1. The authors report the identification of conserved and divergent molecular markers across multiple species with RNA-Seq, but they do not validate the expression of novel markers in either category with an independent method (e.g. in situ or antibody staining).Such validation is needed to substantiate the authors' conclusions.

We have added RNA in situ hybridization results in Figure 3C and Figure 3—figure supplement 1 and 2. Most of the genes were expressed in tissues in accordance with the sequencing results (6 out of 9 common MN genes; 4 out of 6 mouse specific genes; 5 out of 7 skate specific genes). Specifcally, Uchl1, Slc5a7, Alcam, and Serinc1 are expressed in MNs of all three species; Coch, Ppp1rc, Ctxn1, and Clmp are expressed in MNs of mouse but not in MNs of other species; Eya1, Etv5, Dnmbp, and Spint1 are expressed in MNs of skate but not in MNs of other species. In the result section, we have summarized the results as follow:

“These results were validated by performing RNA in situ hybridization in tissue sections on a subset of species-specific genes …”

2. The reviewers do recognize that functional analyses are not feasible in the little skate. However, to substantiate the claim that more complex regulatory mechanisms have evolved in tetrapods to accommodate sophisticated motor behaviors, the authors should conduct additional gene expression experiments. For example, as pointed out by reviewer 1, the authors can focus on Foxp1 and Snail.

We have added further discussion and data about differential expression of Foxp1 in mouse and little skate in Figure 5—figure supplement 16 and have discussed as follows: “Foxp1, the major limb/fin MN determinant appears to be differentially regulated in tetrapod and little skate. Although Foxp1 is expressed in and required for the specification of all limb MNs in tetrapods, Foxp1 is downregulated in Pea3 positive MN pools during maturation in mice (Catela et al., 2016; Dasen et al., 2008). In addition, preganglionic motor column neurons (PGC MNs) in the thoracic spinal cord of mouse and chick express half the level of Foxp1 expression than limb MNs. Although PGC neurons have not yet been identified in little skate, we tested the expression level of Foxp1 using a previously characterized tetrapod PGC marker, pSmad. We observed that Foxp1 is not expressed in MNs that express pSmad (Figure 5‒figure supplement 3). Since there is currently no known marker for PGC MNs in little skate, our conclusion should be taken with caution.”

As for Snai1, in the revision we performed a motif enrichment analysis with an unbiased gene list where Snai1 didn’t show up. However, when we performed an RNA in situ hybridization experiment for Snai1 (Figure 5—figure supplement 3), we found that Snai1 is expressed in MNs of both mouse and little skate, but not in chick, which has been shown previously (Cheung et al., 2005). In order to examine the function of Snai1 in the regulation of Foxp1 expression, we ectopically expressed Snai1 in chick spinal cord by performing in ovo electroporation. However, we did not detect any changes in Foxp1. Instead we observed an increase in the number of neurons and abnormal MN exits from the spinal cord, which is the reminiscent of a previous observation (Zander et al., 2014). Although we did not detect any changes in Foxp1 expression, we cannot rule out the possibility that Snai1 regulates Foxp1 in mouse and little skate, which may require a gene knock out experiment. Because binding sites of Snai1 were not enriched in the new gene sets that we analyzed in the revision, we have not further discussed the Snai1 in the text.

3. Reviewers 1 and 3 raised a number of important issues regarding the bioinformatic comparisons (e.g., completeness of datasets, quality control, introduction of bias) that must be carefully addressed. First, the authors must show validation of their RNA-Seq method in chick, as suggested by reviewer 1. Second, the reviewers would like to see a re-analysis of the RNA-Seq datasets in an unbiased manner, for example, by directly comparing MN expression of orthologous genes in the different species (e.g., compare highly expressed genes in MNs of the different species). Two reviewers stressed that comparing a pure MN population to tail tissue in the skate and DRGs in the mouse was not an appropriate comparison. Third, comparisons of mouse and skate-accessible chromatin regions should be done without biasing first to DEX genes. Lastly, additional key aspects of evolution, such as paralogue substitution or expression of species-specific genes should also be considered. Authors could follow comparisons similar to: 10.1038/s41559-021-01580-3.

Regarding the four suggestions, additional experiments and analyses have been added.

1) An additional experiment was conducted on the marker genes of chicken MN to support RNA-seq data and has been added to the Figure 3—figure supplement 1. In summary 11 out of 11 genes showed highly enriched expression pattern in MNs. The gene list includes Galm2, Cap43, str5, Scn2A, Rtn1, Pcp4, Tgs7bp, Mdga1, ChrnB3, Gpc3, and Hs6st3. In the result section, we have added as follow:

“…wing level MNs of chick embryos at Hamburger-Hamilton (HH) stage 26–27, which was validated by RNA in situ hybridization (Figure 3—figure supplement 1).”

2) The RNA-seq data has been extensively re-analyzed to address the concerns from the reviewers. Instead of using data from tail-SC of skate and sensory neuron tissues of mouse, comparison was made among the homologous MN populations, and Figures 3 and 5 have been updated accordingly.

3) The comparison between mouse and skate ACRs was reanalyzed without biasing to DEGs. The entire gene set and the highly expressed genes (expression percentile > 70) were used in the new analysis to avoid biasing genes (Figure 5 and Figure 5—figure supplement 1). Figure 5 has been revised following this new result.

4) As the reviewer mentioned, our analysis (Figure 3) could not compare genes which do not have one-to-one orthologues. To account for this, additional analysis was conducted on paralog genes. The eggNog program was used to identify paralogs, and species-specific genes whose functions could be substituted by paralogues are discussed (Figure 3—figure supplement 3, Additional Data 5 and 6). Also, species-specific ortholog groups missed on the analyses with the ortholog genes (Figure 3) were identified.

4. The reviewers agree that a significantly improved version of the little skate genome is an important contribution. In the first part of the manuscript, the authors should mention the work by Marletaz et al. (bioRxiv) that also provides a new little skate sequenced genome.

We have referenced the study by Marletaz in the introduction as follows:

“In response to the limited reference genome of little skate, this and a similar study by Marletaz et al., generated new genome assemblies. Marletaz et al. generated a new chromosome-scale genome via combination of Pacbio, Illumina and Hi-C sequencing.”

5. There is a propensity to overstate claims throughout the manuscript. For example, without functional data in little skate, the claim that the TF networks are much simpler is not substantiated simply by ATACseq and predicted binding sites. The authors should tone down statements in Abstract and Discussion (as indicated by the reviewers) and always make it very clear that they are simply predicting regulatory connections between all these genes.

We have rephrased some of the misleading statements to tone them down as suggested by the reviewers.

Reviewer #1 (Recommendations for the authors):1) Based on the low convergence of mouse, skate and chick MN transcriptomes it may be a possibility that one of the datasets is incomplete. The authors should go back to their RNAseq datasets and check for the expression of known MN markers that are conserved in the 3 species to identify where this inconsistency arises. It would also be helpful to show validation of their chick RNAseq method, for example, show the efficiency and specificity of GFP expression in MNs, and what % of MNs were electroporated and sorted before sequencing.

We have validated the chick RNA sequencing by performing RNA in situ hybridization. The results are shown in Figure 3—figure supplement 1. Out of 11 tested genes, 11 genes were expressed in MNs. We have also added details of chick electroporation experiment as follow:

“After electroporation only well electroporated embryos ( > 50% of the brachial MNs) were selected and used for cell sorting. During the FACS sorting, the YFP positive cells were counted to be around 5–10% of the entire population, shown in Figure 3‒figure supplement 1C. After the FACS sorting, we confirmed under fluorescence microscope that most of the sorted cells (> 90%) were YFP positive.”

2) Another surprising finding is that a higher number of putative TF binding sites are found in the tail-SC-specific ACRs for pectoral MNs and that the expression levels of these TFs do not correlate with this trend. The authors should comment on this discrepancy. Similarly, while they were found to be highly expressed in pectoral MNs, 5 genes including Isl1, show no associated ACR (Figure 5C). Do the authors think that this is due to long-range regulation of those genes? Some discussion of this would be clarifying.

We have addressed these issues in the Discussion as follows:

“In addition, whether the predicted TFs actually bind to the putative binding sites and whether TFs function as activators or repressors are yet to be validated.”

We only consider ACRs in the region of 10 kb upstream and 10 kb downstream from the genes, which may exclude the long-range enhancers. As reviewer #2 suggested, we discussed this issue as follows:

“As another limitation of the current study, the ACRs analyzed here did not consider longrange regulatory elements, which might have caused the lack of associated ACRs in the MN genes (Figure 5C). For more accurate characterization of regulatory elements, future studies could provide a more comprehensive approach including ChIP-seq and chromatin conformation capture assay together with functional validation of the binding sites.*”*

Reviewer #2 (Recommendations for the authors):As already stated, the main limitation of the study seems to be the strategy followed for the RNAseq and ATACseq comparative analysis. The use of a prior intra-species comparison with heterogeneous criteria is not well-justified, it is a potential source of bias and should be avoided. Why authors do not directly compare MN expression and ATACseq of orthologous genes in the different species? In addition, it is known that there are extensive paralogue substitutions that indicate common regulatory mechanisms but might be missed if not analysed directly. Authors could follow comparisons similar to previously published articles: 10.1038/s41559-021-01580-3.

As reviewer #2 suggested, we have extensively re-analyzed the RNA-seq and ATAC-seq data. To avoid biases caused by heterogeneous cell type data, RNA sequencing data was generated from forelimb MNs of mouse and the new analysis focuses on comparisons of homologous MN populations between different species (Figure 3). In addition, the ATACseq data was re-analyzed while avoiding biasing the genes to statistically significant DEGs. The new analysis use all genes or highly expressed genes (Figure 4, 5 and Figure 5‒—figure supplement 1). We agree that there are extensive paralogue substitutions in the homologous cell types during evolution. To account for those genes without one-to-one orthologs, which could also be important for creating species-specific differences, we have added paralog gene analysis using eggNOG. The species-specific genes whose functions could be substituted by paralogues (Figure 3—figure supplement 3, Additional Data 5 and 6). Also, species-specific ortholog groups missed out on the analyses with the ortholog genes (Figure 3) were identified. We have summarized the results as follows:

“However, considering 11, 17 and 4 genes of mouse, skate and chick MN-specific genes are paralogs of one of the shared genes (Figure 3—figure supplement 3B), the degree of conservation is remarkable between MNs of these species while there are only a few remaining genes which could be species-specific. In addition to the comparisons among ortholog genes, we also identified 12, 171 and 6 paralog genes specifically expressed by skate, mouse and chicken MNs, respectively which could further explain species-specific differences of MNs (Figure 3—figure supplement 3B).”

Comments on each section:1) Abstract:Several statements are not strongly supported by data:"Conserved MN genes were enriched for early-stage nervous system development.""conservation of the potential regulators with divergent transcription factor (TF) networks through which expression of MN genes is differentially regulated.""TF networks in little skate MNs are much simpler than those in mouse MNs."

The statements have been toned down following the reviewer #2’s comment. The abstract is rephrased as follows:

“Through interspecies comparison of mouse, skate and chicken MN transcriptomes, shared and divergent MN expression profiles were identified. Comparison of accessible chromatin regions between mouse and skate MNs predicted shared regulators with divergent transcription factor (TF) binding sites, which could be used for achieving differential regulation of MN-restricted genes. Predicted number of TF binding sites in the genic region in mouse MNs were much larger than those in the little skate MNs.”

2) Figure 1 data:In the first part of the manuscript, the authors should mention biorxiv Marletaz et al.'s work which also provides a new little skate sequenced genome and analysis, with a higher number of identified coding genes compared to this work and also with 3D chromatin structure data that identifies TAD boundaries that coincide with CTCF sites in Hoxa and Hoxd clusters mentioned in Baek et al.

We have addressed this comment in the essential revisions #4.

3) Figure 2 data:(3.1) The comparison of Pectoral MN with tail SC, compares a purified population with a mix of cell types including MN. This unbalanced comparison could explain why tail SC shows 3 times more DEX genes than pec-MN.

We agree with reviewer #2. In the revision, we have addressed this issue in the Results as follow:

“Larger number of DEGs in tail-SC may be caused by comparing heterogeneous cell types in tail-SC with homogeneous cell type in pec-MNs.”

(3.2) In the previous analysis published by the group, 592 genes were identified as 2 fold enriched in pectoral MN, why the new analysis retrieves only 135 genes? This should be clarified for the reader.

As reviewer #2 suggested, we have clarified this in the results as follows:

“Although the total number of DEGs are different from the previous data (592 vs. 135 genes in pec-MN DEGs), which might be caused by different statistical analysis with different reference genome…”

(3.3) Also from previous work of the authors several genes were determined to be expressed in little skate MN (HB9, lhx3, lhx1, Ephs, Ephrins, slit3, onecut1, nell2, nrcam, ngfr, pappa2, cdh7, amigo1, unc5c, lrx5, lrp1b, Zfp804a, etc), which of these genes are predicted to be enriched (or expressed at least) in the Pec-MN data set? Were all the genes found? This could be used as quality control of the data set or data set analysis.

We have compared the re-analysis result with the previous RNA seq result in Figure2‒ figure supplement 1 and added this information in the Results as follows:

“… previous RNA-seq data based on de novo assembly and annotation using zebrafish was mostly recapitulated in our DEG analysis based on our new skate genome (21 out of 24 previous fin MN marker genes have the expression level ranked above 70^th^ percentile in PecMNs; Figure 2‒figure supplement 1).”

4) Figure 3 data:(4.1) Compares "Highly expressed genes in pec-MN". What are the criteria followed to assigned "highly expressed genes"?

In the revision, we have defined highly expressed genes as genes that have the expression levels ranked above 70^th^ percentile in each species.

(4.2) This data set is now compared with DEX of mouse embryonic MN compared to sensory neurons. Which type of mouse MN? at what A-P level?

In order to compare homologous cell populations from different species, in the revision we newly generated RNA sequencing data of forelimb MNs in mouse embryo at e13.5, which are the homologous population of pec-MNs in skate. The sequencing results are confirmed by RNA in situ hybridization (Figure 3; Figure 3—figure supplement 2). The RNA sequencing data with sensory neuros were not used in the revision. In the result section, we have added as follows:

“In order to compare gene expression with homologous cell types from each species, we performed RNA sequencing experiments with forelimb MNs of mouse embryos at embryonic day 13.5 (e13.5)”

(4.3) If skate genes are selected by high expression, why do mouse MN not follow the same criteria to avoid bias of comparison? As a control for bias in the criteria: What is the overlap of mouse highly expressed genes (with similar criteria to skate gene selection) with the data set of mouse MN DEX compared to sensory mouse neuron?

In the revision, we have defined highly expressed genes as genes that have the expression levels ranked 70^th^ percentile in each species.

(4.4) Same for chick data, how equivalent are brachial MN to mouse MN dataset? Why select DEX compared to the brachial spinal cord and not sensory neurons as in mouse.

As reviewer suggested, in the revision we have generated new RNA sequencing data with forelimb MNs in mouse and compared gene expression in homologous cell types in the different species (forelimb MNs in mouse; br-MNs in chicken; pec-MNs in little skate).

Without any justification for the followed strategy, it seems different populations and comparisons will introduce important biases that limit the conclusions.The best strategy would be to compare highly expressed genes in MN in the different species and clarify if used populations are really "homologous" cell types in the different species.

As reviewer #2 suggested, we have revised the new RNA-seq analysis to show the comparisons of highly expressed genes ( > 70 percentile) in MNs. The tail-SC, ce-SC and sensory neurons which could complicate and bias the RNA-seq analysis are no longer used as the controls for interspecies comparison.

(4.5) The variability of sampled populations or the biases introduced in the analysis might explain the low overlap found among the 3 species (Figure 3A). For the Venn diagram, is the overlap among categories higher/lower/equal to what would be expected by chance?

In the revision we have compared gene expression from the MNs from the similar region in the different species. The Venn diagram was changed to include new analysis results where there are now 1038 commonly expressed genes in MNs of three species.

5) Figure 4 data:(5.1) Similar to previous comments: Why limit TF motif enrichment of ACRs to only DEXs in pec-MN and tail-SC that (1) highly reduced the number of ACRs and (2) Induces a bias in the analysis? Analysis should be performed with all MN ACRs.

We agree with the reviewer #2. In the revision, we have re-analyzed the ATAC-seq data with all the ACRs or ACRs in highly expressed genes (> 70 percentile). The ATAC-seq results have been updated accordingly in the Figure 4, 5 and Figure 5—figure supplement 1.

(5.2) Line 224: On the other hand, the tail-SC-specific ACRs in the pec-MN DEGs were enriched with the binding sites of Hoxa9, Hoxa11, Hoxd10 and Hoxd11, most of which are expressed in fin-MNs of little skate.Could the authors provide a sentence to try to explain this observation? I found confusing that this is mentioned without providing more context.

We meant to suggest that DEG is not only generated by gene activation but also gene repression; tail-SC Hox genes are predicted to bind to genes enriched in pec-MNs and this binding may lead to repression of the genes in tail-SC. In the revision, we restated this part and the Limitation as follows:

“Fin-MN-specific ACRs were enriched with predicted binding sites of Hoxa5 and Hoxd9, which are expressed in the fin-MNs of little skate, while the tail-SC-specific ACRs in the pecMN genes (percentile > 70) were enriched with the predicted binding sites of Hoxd11 and Hoxa13, expressed in tail SCs of little skate.”

“…whether the predicted TFs actually bind to the putative binding sites and whether TFs function as activators or repressors are yet to be validated.”

6) Figure 5 data:(6.1) Similar to previous comments, conclusions driven by this analysis might be biased by wrong comparative strategies to start with, thus conclusions are not well supported. Comparative of mouse and skate ACRs should be done without biasing first to DEX genes, which highly reduced the number of analysed ACRs, particularly to shared genes that are only 40.

As with the response to the reviewer #2-5.1 comment, in the revision, we have re-analyzed the ATAC-seq data with all the ACRs or ACRs in highly expressed genes (> 70 percentile). The ATAC-seq results have been updated accordingly in the Figure 4, 5 and Figure 5—figure supplement 1.

(6.2) MN TF enrich motifs are broadly found in mouse sensory and skate tail DEX genes, which might be a strong indication that the strategy used in the analysis is not suitable for the identification of MN gene regulatory networks.

As with the response to the reviewer #2-5.1 comment, in the revision, we have re-analyzed the ATAC-seq data with all the ACRs or ACRs in highly expressed genes (> 70 percentile). The ATAC-seq results have been updated accordingly in the Figure 4, 5 and Figure 5—figure supplement 1.

(6.3) Figure 5C: Most TF binding motifs in databases are built with experimental data from mouse or human TFs. Although core sequences for TF binding sites in the same families are conserved, small differences arise in individual members or between species. Could this explain why there is a higher number of motifs found in mouse than in skate? In addition, is the number of motifs in mouse and skate normalized by the size of analysed sequence?

We agree with the reviewer’s concern. In the revision, we have included the limitation of the analysis as follows:

“However, it is important to note that the motif analysis may be biased towards mouse data, because our motif enrichment analysis is based on motif data from mouse TFs.”

The enrichment of motifs in Figure 5, which uses HOMER, is generated with considering the size of analyzed sequences.

7) General comments on format:(7.1) Figures 3-5: differences among circles sizes are difficult to appreciate, maybe color heatmaps would be more informative.

The figures have been modified accordingly in response to the reviewers’ comment.

(7.2) Figure 4 B, C could be included in supplementary.

Thanks for the suggestion. The figure 4B and C quantifies the locations of ACRs that are assigned to genic regions. The genic region is used for annotating genes that each ACR is associated with, so we think this is an important piece of information to follow the manuscript and have decided to keep the 4B and C in Figure 4.

Reviewer #3 (Recommendations for the authors):The authors should be careful in their phrasing so as to always make it clear that they are simply predicting regulatory connections between all these genes.For instance, on Page 3, line 29:"Comparison of accessible chromatin regions between mouse and skate MNs revealed conservation of the potential regulators with divergent transcription factor (TF) networks through which expression of MN genes is differentially regulated. TF networks in little skate MNs are much simpler than those in mouse MNs, suggesting a more fine-grained control of gene expression operates in mouse MNs."As one can see, without functional data in little skate, the claim that the TF networks are much simpler is not substantiated simply by ATACseq and predicted binding sites.A similar claim on Page 17, line 338 indicates the same propensity to overstate these claims:"As illustrated in the Figure 5C, a greater number of shared TFs binds to their downstream TFs in mouse MNs than in skate pec-MNs, allowing an intricate control of gene expression and thus the more complex nervous system of mouse compared to that of skate"Again, there was no demonstration of little skate TFs binding to any downstream TF genes, only predicted binding sites. Thus, the claim is unsubstantiated.These examples indicate an overall propensity by the authors to gloss over the fact that no functional connections between any TFs or genes were experimentally demonstrated in little skate. I do not believe it is appropriate to substitute binding motif enrichment for experimentally determined regulatory connections when inferring gene regulatory networks in something as complex as vertebrate development. There are a number of reasons why this is problematic.

As reviewer suggested, we have revised the manuscript to avoid overstatements and we have discussed the limitations as follows:

“However, it is important to note that the motif analysis may be biased towards mouse data, because our motif enrichment analysis is based on motif data from mouse TFs. In addition, whether the predicted TFs actually bind to the putative binding sites and whether TFs function as activators or repressors are yet to be validated. As another limitation of the current study, the ACRs analyzed here did not consider long-range regulatory elements, which might have caused the lack of associated ACRs in the MN genes (Figure 5C). For more accurate characterization of regulatory elements, future studies could provide a more comprehensive approach including ChIP-seq and chromatin conformation capture assay together with functional validation of the binding sites.” [Editors’ note: further revisions were suggested prior to acceptance, as described below.]

Essential revisions (please note that no additional experiments are required at this point):1. Please clarify whether you performed ATAC-seq from mouse MNs.

In the previous version of the manuscript, we analyzed ATAC seq data from MNs of entire spinal cord (Closser et al.). As reviewer #1 and #3 suggested, we used a more recent ATAC seq dataset acquired specifically from forelimb and hindlimb level MNs (Sawai et al., 2022), to match the cell type as close as possible to skate fin MNs. In the revised version, the source of the ATAC seq data have been referenced as follows:

“…the TF motif prediction was performed with the ACRs of skate fin MN compared with the mouse limb-level MN ATAC-seq data (34)…”

“…ACRs of genes expressed in cortical PV interneuron and excitatory neurons (35) …”

2. The findings on Snail1 are not described, and this part should be removed.

We have removed the *Snai1* data.

3. Please address the concerns raised by reviewer #3 about Figure 5.

The concerns on cell types used:

To address the concerns raised by Reviewer #3, we have reanalyzed data in Figure 5 by using publicly available ATAC data of limb-level MNs (forelimb and hindlimb MNs). At limb levels, most of the MNs are of the LMC subtype, so we believe this very closely matches the compared cell types (limb MNs of mouse and fin MNs of skate). From the analysis we found more TF motif predictions in ACRs of genes that are expressed in mouse MNs than in skate MNs. As reviewer suggested, we have mentioned the caveat of this finding in the results as follows:

“However, it is important to note that the larger number of TF motif predictions in mouse MN-expressed genes could be due to the biased motif database toward mouse TFs.”

The concerns about the meaning of the comparative motif prediction analysis:

Among the TF motif predictions, 14 TF motif predictions were commonly found in genes expressed in skate fin-MNs and mouse limb-level MNs. However, only a subset (10 TF motifs in cortical PV interneurons: Sp5, Sp2, Smad3, Rfx6, Rf1, Bmyb, Klf5, Klf3, Lhx1, En1; 8 TF motifs in cortical excitatory interneurons: Sp5, Sp2, Smad3, Rfx6, Rf1, Bmyb, Klf5, Klf3) of the 14 TF motifs were enriched in ACRs of genes expressed in cortical PV interneurons and excitatory interneurons in mouse. We think this suggests that the 14 TF motifs would not be randomly observed and that the 14 TF motifs may have functions in regulating cell types or regional identities of neurons.

4. We acknowledge that the revised version has toned down most claims on gene regulatory mechanisms, but we further suggest toning down the claim made in the title of the paper.

We have edited the title as follows: “Little skate genome provides insights into genetic programs essential for limb-based locomotion”

5. The manuscript should be edited for language and grammatical errors and carefully proofread.

The language and grammatical errors in the manuscript have been proofread.

Reviewer #1 (Recommendations for the authors):The manuscript is much improved, as the authors have addressed a number of the major previous concerns, including re-analyzing RNA-seq data, generating a new set of mouse RNA-seq data, and validating their results with in situ hybridization. The new analysis seems much more convincing than the previous one. Some lingering concerns remain, mostly regarding figure 5 and the central claim of the paper, which is still not functionally supported. The authors did however tone down their claims and pointed out the limitations of the study.(1) Did the authors perform ATAC-seq from mouse MNs? They show some data in figure 5 but it was not clear from the description of the methods or the results whether they performed the experiment.

In the previous version of the manuscript, we analyzed ATAC seq data from MNs of entire spinal cord (Closser et al.). As reviewer #1 and #3 suggested, we used a more recent ATAC seq dataset acquired specifically from forelimb and hindlimb level MNs (Sawai et al., 2022), to match the cell type as close as possible to skate fin MNs. In the revised version, the source of the ATAC seq data have been referenced as follows:

In the Results,

“…the TF motif prediction was performed with the ACRs of skate fin MN compared with the mouse limb-level MN ATAC-seq data (34)…”

“…ACRs of genes expressed in cortical PV interneuron and excitatory neurons (35) …”

In the Materials and methods,

“The ATAC-seq raw data of mouse limb-level MNs (brachial and lumbar) published under the GEO accession of GSE175503 (34) was used for comparing with skate ATAC-seq data. The mouse cortical PV interneuron and excitatory neuron RNA-seq and ATAC-seq data (35) with GEO accession of GSE63137 were downloaded.”

(2) It seems that repeating the motif analysis of the ATAC-seq data gave completely new regulatory factors for FoxP1, even though the same regions as in the original manuscript are shown. It appears that Snai1 is no longer on the list so the experiment in the supplemental figures testing its function is somewhat moot. The authors also do not describe the experiment at all in the Results section so it should be taken out.

The figure containing the *Snai1* data has been removed from the manuscript.

Reviewer #3 (Recommendations for the authors):The revised manuscript has made important efforts in answering some of the raised concerns but did not address other key issues, particularly at the end of the manuscript.The description of the little skate genome is, with no doubt, useful for the field of Evolutionary Biology. Then authors, re-analyse previously published experimental data and compare differential gene expression between isolated pectoral motorneurons and a mixed population of the tail spinal cord of little skate (Figure 2) and differential accessible chromatin regions between Fin motorneurons (Pectoral+pelvic) compared to a mixed population of the tail spinal cord (Figure 4). These two figures are descriptive and it is unclear what insights are provided by the comparison of these heterogeneous populations of cells. Thus do not seem to constitute a great advancement in the understanding of gene regulatory networks present in the little skate or their evolution.In the new version of the manuscript authors now compared homologous motorneuron populations of the little skate, mouse, and chick, which reveals a remarkable degree of conservation of the transcriptomes in the 3 species (Figure 3), which is even higher when considering possible paralog substitutions in the few divergently expressed genes. This is radically different from the initial version of the manuscript and I think is an important observation.Finally, the authors attempt to describe and compare motorneuron gene regulatory networks in mouse and skate in Figure 5. I think this is the weakest part of the manuscript, not only I found it hard to follow, not clearly explained, and using vague terms and overstatements but in addition the methodology and strategy for the analysis seem incorrect, leading to potentially wrong conclusions.To start, it is unclear to me what is exactly the mouse motorneuron population that is being compared and how equivalent it is to the mixed Pectoral and pelvic motorneuron population from the little skate.

To address the concerns raised by Reviewer #3, we have reanalyzed data in Figure 5 by using publicly available ATAC data of limb-level MNs (forelimb and hindlimb MNs; Sawai et al.). At the limb levels most of the MNs are of the LMC subtype, so we believe this would very closely match the cell types of comparison (limb MNs of mouse and fin MNs of skate).

More importantly, Figure 5A: authors describe 18 TF motifs commonly enriched in mouse and skate MN ACRs, among which 6 TFs predicted to bind those motifs are expressed both in mouse and skate. What does this mean? Is this overlap biologically meaningful? How many would overlap if enriched motifs in skate ACRs are compared to motifs in mouse ACRs of a non-related neuron type at a similar developmental stage, such as cortical interneurons cortical projection neurons? Less? More? similar number?

Among the TF motif predictions, 14 TF motif predictions were commonly found in genes expressed in skate fin-MNs and mouse limb-level MNs. However, only a subset (10 TF motifs in cortical PV interneurons: *Sp5*, *Sp2*, *Smad3*, *Rfx6*, *Rf1*, *Bmyb*, *Klf5*, *Klf3*, *Lhx1*, *En1*; 8 TF motifs in cortical excitatory interneurons: *Sp5*, *Sp2*, *Smad3*, *Rfx6*, *Rf1*, *Bmyb*, *Klf5*, *Klf3*) of the 14 TF motifs were enriched in ACRs of genes expressed in cortical PV interneurons and excitatory interneurons in mouse. We think this suggests that the 14 TF motifs would not be randomly observed and that the 14 TF motifs may have functions in regulating cell types or regional identities of neurons.

Authors then look for motif enrichment of these 6 common motif/TFs in ACRs of other expressed TFs (either expressed both in mouse and skate or only in one of them). It is unclear why TFs with enriched motifs in mouse and skate should be upstream of the other TFs, it is equally possible that the 6 common motif/TFs are downstream of the expressed TFs. In addition, using the term "interactions" for motif enrichment of one TF in the ACRs of another TF is an overstatement.

We understand the reviewer #3’s concern. In the revised manuscript, this analysis has been removed. And the term “interaction” has been edited to avoid overstatement.

It is also striking that "interactions" of the 6 mouse common motif/TFs seem equally prevalent within ACRs of mouse-expressed TFs compared to ACRs of TFs expressed in skate but not in mouse. Similarly, for the skate data, motif enrichment in ACRs of commonly expressed TFs or TFs only expressed in the skate is not higher than in ACRs of TF expressed in mouse only. These results again raise doubts if this analysis is meaningful at all. Would it be different using just randomly picked ACRs from other TFs not expressed in MN? Finally, as already mentioned in the first review, the increasing number of motifs for mouse TFs could be again due to methodological limitations (based on known motifs in mouse compared to skate, etc) but not biologically meaningful, it should be stated right away when described and not as a separate paragraph in the section of limitations for the study.

As reviewer #3 suggested, we have mentioned the caveat of this finding in the results as follows:

“However, it is important to note that the larger number of TF motif predictions in mouse MN-expressed genes could be due to the biased motif database toward mouse TFs.”

Considering all these methodologic concerns, together with the lack of any experimental validation of predicted binding sites for any of the TFs, it would be advisable to completely remove that figure.In summary, the manuscript has valuable information, namely the description of the little skate genome and the comparison of mouse, chick, and little skate MN transcriptome. It also provides a description and comparison of some heterogeneous populations of neurons in little skate which provide limited insights into gene regulation in skate motorneurons. Most importantly, I think the last part of the manuscript (Figure 5) is below the standards required for a journal such eLife. Unfortunately, I don´t think the paper, in its current revised format, provides any mechanism underlying the evolution of vertebrate locomotion as stated in the title.Other comments:Authors constantly use the term "MN genes" or similar expressions when should be referring to "MN expressed genes" or "MN enriched genes". Other non-accurate expressions:"Number of predicted TFs enriched" when it refers to TF motif predictions, "expression of each predicted TF" when meaning "TF expression for the corresponding TF enriched motif ", "predicted shared TFs found in MN ACRs" meaning " predicted shared enriched TF motifs found in MN ACRs", etc.This lack of accuracy makes it harder for the reader to follow the text.

The inaccurate terms have been removed or modified throughout the manuscript.

Methods include snail cDNA electroporation, but I don't think those results are included in the manuscript.

The methods describing *Snai1* cDNA electroporation have been removed from the manuscript.

In figure 4D, the legend states ACR group but would be clearer to label as Motif category or Motif distribution in ACR categories.

The legend in Figure 4D have been edited to “Motif category”.

Line 251: "Together with the predicted binding sites of the Hox proteins, the binding sites of Foxp1 and Pbx1 and Lhx3, which are well-known regulators in MN(31-33), were enriched in shared ACRs and fin-MN-specific and tail-SC-specific ACRs, respectively (Figure 4D)"However in the figure Pbx1 is also in the shared category as Foxp1, not in fin-MN specific, correct?

The statement has been corrected as follows:

“…motif predictions of Foxp1, Pbx1, and Lhx3, well-known regulators in MN(31-33), were found in the ACRs; the motif predictions of Foxp1and Pbx1 in shared ACRs and Lhx3 in fin-MN-specific and tail-SC-specific ACRs, respectively…”

Why Figure 5 shows common TFs only 16 TFs, if figure 3 shows many more?

In the previous Figure 5 we analyzed only the 16 TFs that are highly expressed in MNs but not in tail SC or cervical SC. However, in the revised manuscript the network plot showing these TFs have been removed from the manuscript to avoid misunderstanding.